# VCR-Bench: A Comprehensive Evaluation Framework for Video Chain-of-Thought Reasoning

## Abstract

The advancement of Chain-of-Thought (CoT) reasoning has significantly enhanced the capabilities of large language models (LLMs) and large vision-language models (LVLMs). However, a rigorous evaluation framework for video CoT reasoning remains absent. Current video benchmarks fail to adequately assess the reasoning process and expose whether failures stem from deficiencies in perception or reasoning capabilities. Therefore, we introduce **VCR-Bench**, a novel benchmark designed to comprehensively evaluate LVLMs' **V**ideo **C**hain-of-Thought **R**easoning capabilities. VCR-Bench comprises 859 videos spanning a variety of video content and durations, along with 1,034 high-quality question-answer pairs. Each pair is manually annotated with a stepwise CoT rationale, where every step is tagged to indicate its association with the perception or reasoning capabilities. Furthermore, we design seven distinct task dimensions and propose the CoT score to assess the entire CoT process based on the stepwise tagged CoT rationals. Extensive experiments on VCR-Bench highlight substantial limitations in current LVLMs. Even the top-performing model, o1, only achieves a 62.8% CoT score and an 56.7% accuracy, while most models score below 40%. Experiments show most models score lower on perception than reasoning steps, revealing LVLMs' key bottleneck in temporal-spatial information processing for complex video reasoning. A robust positive correlation between the CoT score and accuracy confirms the validity of our evaluation framework and underscores the critical role of CoT reasoning in solving complex video reasoning tasks. We hope VCR-Bench to serve as a standardized evaluation framework and expose the actual drawbacks in complex video reasoning task.

## 1 Introduction

The emergence of Chain-of-Thought (CoT) reasoning (Wei et al., 2022) has significantly enhanced the reasoning capability of large language models (LLMs), as evidenced by the recent breakthroughs of DeepSeek-R1 (Guo et al., 2025a) and OpenAI o1 (OpenAI, 2024a). By generating human-like, interpretable reasoning steps, these reasoning models have demonstrated remarkable advantages in solving complex visual tasks. Recently, large vision-language models (LVLMs) (OpenAI, 2024b; Chen et al., 2024a;b;c) have achieved groundbreaking progress in multiple visual fields, especially in research on CoT reasoning for video data.

However, video understanding field still lacks a scientifically effective evaluation suit for CoT reasoning, with existing benchmarks primarily suffering from the following two shortcomings: First, current video benchmarks (Xu et al., 2017; Liu et al., 2024; Zhou et al., 2024; Zhao et al., 2025b) often lack comprehensive annotations of CoT steps, focusing only on the accuracy of final answers during model evaluation while neglecting the quality of the reasoning process. This evaluation approach makes it difficult to comprehensively evaluate model's actual drawbacks during the CoT reasoning process. As shown in Figure 1, the model captures one piece of erroneous information while missing one correct piece during its reasoning process, yet ultimately arrives at the correct final answer. Second, existing video understanding benchmarks (Li et al., 2023; Fu et al., 2024) fail to effectively distinguish performance differences in perception and reasoning capabilities. The

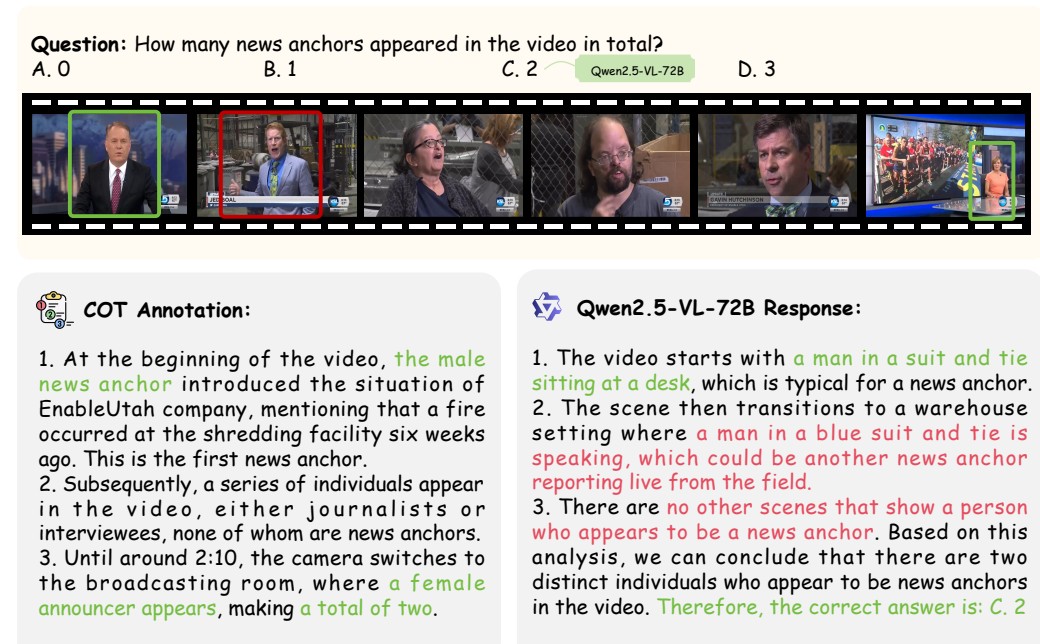

Figure 1: **Failure case of accuracy-based evaluation.** The video contains two news anchors, but the model missed one while misclassify a non-anchor as an anchor, yet reached the correct answer. This suggests that relying solely on accuracy is insufficient for appropriately evaluating a model's performance under video CoT reasoning.

absence of an effective evaluation suit has become a significant bottleneck that hinders the in-depth development of complex reasoning research in the field of video understanding.

To fill this gap, we propose **VCR-Bench**, a benchmark specifically designed to evaluate the **V**ideo **C**hain-of-Thought **R**easoning capabilities of LVLMs. We have constructed a multi-dimensional evaluation framework, defining seven distinct task dimensions that comprehensively cover a diverse range of video types and durations. For each data sample, in addition to providing a standard answer, we have meticulously curated detailed and accurate reference stepwise rationals as CoT annotation. All samples underwent rigorous manual annotation and quality control, ultimately resulting in the creation of VCR-Bench, which includes 859 videos and 1,034 high-quality question-answer pairs. We draw on existing work in the field of image understanding (Jiang et al., 2025; Chen et al., 2024d; Thawakar et al., 2025) to innovatively design an evaluation framework specifically for assessing generated CoT reasoning steps. This framework first categorizes the CoT steps into visual perception steps and logical reasoning steps, then systematically evaluates the CoT steps across multiple dimensions including recall rate and precision rate to derive the CoT score, thereby providing a basis for comprehensively measuring models' reasoning capabilities.

We thoroughly evaluated multiple models on VCR-Bench. Results reveal significant limitations: even the top model, o1 (OpenAI, 2024a), achieves only 62.8% CoT score and 56.7% accuracy, while most models score below 40%. The gap highlights LVLMs' shortcomings in video reasoning and the need for improvement. Lower perception scores compared to reasoning scores indicate that temporal-spatial understanding remains the main bottleneck. Further analysis shows a strong positive correlation between CoT scores and accuracy, validating the reliability of our evaluation framework.

In a nutshell, our core contributions are as follows:

- To our knowledge, VCR-Bench is the first benchmark specifically designed for video CoT reasoning. Through rigorous manual annotation, we provide detailed reasoning steps for each sample, ensuring data accuracy and reliability while offering the research community a high-quality video reasoning evaluation benchmark.

Figure 2: **Video source and categories.**

Table 1: **Key Statistics of VCR-Bench.**

| Statistic | Number |
|---|---|
| Total Videos | 859 |
| - Short Videos ($\leq$ 1 min) | 418 (48.7%) |
| - Medium Videos ($1 \sim 5$ min) | 293 (34.1%) |
| - Long Videos ($>$ 5 min) | 148 (17.2%) |
| Total Questions | 1034 |
| - Dimensions | |
| Fundamental Temporal Reasoning | 159 (15.4%) |
| Video Temporal Counting | 161 (15.6%) |
| Video Temporal Grounding | 143 (13.8%) |
| Video Knowledge Reasoning | 153 (14.8%) |
| Temporal Spatial Reasoning | 135 (13.1%) |
| Video Plot Analysis | 139 (13.4%) |
| Temporal Spatial Grounding | 144 (13.9%) |
| - Types | |
| Multiple-choice | 510 (49.3%) |
| Open-ended | 524 (50.7%) |
| Total Reference Reasoning Steps | 4078 |
| - Visual Perception Steps | 2789 (68.4%) |
| - Logical Reasoning Steps | 1289 (31.6%) |
| Reasoning Steps per Sample (avg/max) | 3.9/12 |
| Reasoning Step Word Count (avg/max) | 27.0/129 |
| Question Word Count (avg/max) | 22.1/161 |
| Answer Word Count (avg/max) | 3.5/49 |

- We have successfully introduced the CoT evaluation framework into the field of video reasoning, assessing the entire reasoning process based on step-by-step annotated CoT rationales, thereby providing an effective approach to measure the video reasoning performance of LVLMs.

- Through extensive evaluation experiments, we have validated the effectiveness of our assessment methods and data, while also demonstrating that current LVLMs still exhibit significant limitations in video reasoning, especially in the extraction of temporal-spatial information. Furthermore, our experiments demonstrate a strong correlation between CoT step quality and final answer accuracy.

## 2 VCR-BENCH

### 2.1 DATASET CURATION

As shown in Figure 2, to ensure diverse video data and rich sample information, we curated VCR-Bench by integrating multiple existing benchmarks. These include datasets for video perception and comprehension (e.g., Perception Test (Patraucean et al., 2024), NExTVideo (Xiao et al., 2021), TVbench (Cores et al., 2024), MLVU (Zhou et al., 2024), VCGBench-Diverse (Maaz et al., 2024), COIN (Tang et al., 2019)); subject knowledge understanding and reasoning (videoMMMU (Hu et al., 2025), MMVU (Zhao et al., 2025a)); long-form video understanding (Video-MME (Fu et al., 2024), LongVideoBench (Wu et al., 2024)); temporal localization and analysis (ActivityNet Captions (Krishna et al., 2017), ReVOS Videos (Yan et al., 2024)); and video scene reasoning (VideoEspresso (Han et al., 2024)), among others.

#### 2.1.1 TASK DEFINITION

To comprehensively evaluate the differences in LVLMs' capabilities for video Chain-of-Thought (CoT) reasoning from multiple perspectives, we define seven distinct dimensions of task categories. These dimensions encompass various aspects such as spatiotemporal perception, logical reasoning, and knowledge-based analysis. The specific task types are as follows, the specific examples for each dimension can be found in the appendix G.

- **Fundamental Temporal Reasoning (FTR):** FTR task represents a basic temporal reasoning problem, requiring the model to understand temporal order and analyze the sequence of events.

- **Video Temporal Counting (VTC):** VTC task requires the model to calculate the frequency of events or actions and perceive the number of occurrences of specific objects.

- **Video Temporal Grounding (VTG):** VTG task requires the model to locate the specific moment or time interval corresponding to a given action or event.

- **Video Knowledge Reasoning (VKR):** VKR task requires the model to extract knowledge-related information from the video and apply domain-specific reasoning to solve targeted problems.

- **Temporal Spatial Reasoning (TSR):** TSR task focuses on the spatial position changes of characters, including their movement trajectories and specific locations.

- **Video Plot Analysis (VPA):** VPA task requires the model to understand the narrative logic of the video and explain specific events within the plot.

- **Temporal Spatial Grounding (TSG):** TSG task requires the model to locate the spatial position of a corresponding object within a specified temporal sequence.

### 2.1.2 DATA ANNOTATION AND REVIEW

To enable CoT evaluation, we provide questions, answers, and CoT annotations (reference reasoning steps) for all data. These reference steps represent the essential reasoning path to derive correct answers. Our annotation pipeline combines automated generation (using Gemini 2.0 (Pichai et al., 2024)) followed by human verification. This ensures both diversity and accuracy. Each sample's reasoning steps form an ordered set $\mathcal{R} = \{r_1, r_2, ..., r_N\}$ of $N$ atomic sub-steps, designed to facilitate granular evaluation.

### 2.1.3 DATA ANALYSIS

After data annotation and verification, we have ultimately constructed a dataset comprising 859 videos and 1034 question-answer pairs. As shown in Table 1, our video dataset encompasses a wide range of different scenarios, including indoor daily life, sports competitions, outdoor nature, and urban architecture. It covers multiple categories such as personal photography, documentaries, films and television, educational videos, and news reports. The duration of the videos ranges from less than one minute to over 30 minutes, ensuring rich diversity in content and high density of informational cues. Meanwhile, our question-answer pair data achieves a rough balance across seven different dimensions, ensuring the richness and balance of the benchmark tasks.

## 2.2 CoT EVALUATION STRATEGY

Current video understanding benchmarks primarily evaluate the correctness of models' final answers while neglecting intermediate CoT reasoning steps. This evaluation approach fails to provide a comprehensive assessment of models' reasoning capabilities. When addressing complex problems, models must perform multiple cognitive operations including perception and reasoning - evaluating only the final answers cannot reveal their actual shortcomings. As shown in Figure 3, to address this limitation, our proposed VCR-Bench incorporates two additional evaluation components alongside conventional final-answer assessment: CoT Reasoning Deconstruction and CoT Quality Evaluation.

### 2.2.1 CoT REASONING DECONSTRUCTION

The reasoning process of LVLMs involves multiple distinct operations, reflecting diverse capabilities. To systematically evaluate model performance across these competencies, we propose CoT Reasoning Deconstruction, which breaks down the process into two core dimensions:

**Visual Perception** assesses the model's ability to extract spatiotemporal information (*e.g.*, actions, object locations) from videos—the foundational skill for vision tasks.

**Logical Reasoning** evaluates the model's capacity to derive conclusions from perceived information, critical for complex problem-solving.

Formally, we represent reference reasoning steps as: $\mathcal{R} = \mathcal{R}_p \cup \mathcal{R}_r$, where the $\mathcal{R}_p$ and $\mathcal{R}_r$ denote **perception** and **reasoning** subprocesses, respectively.

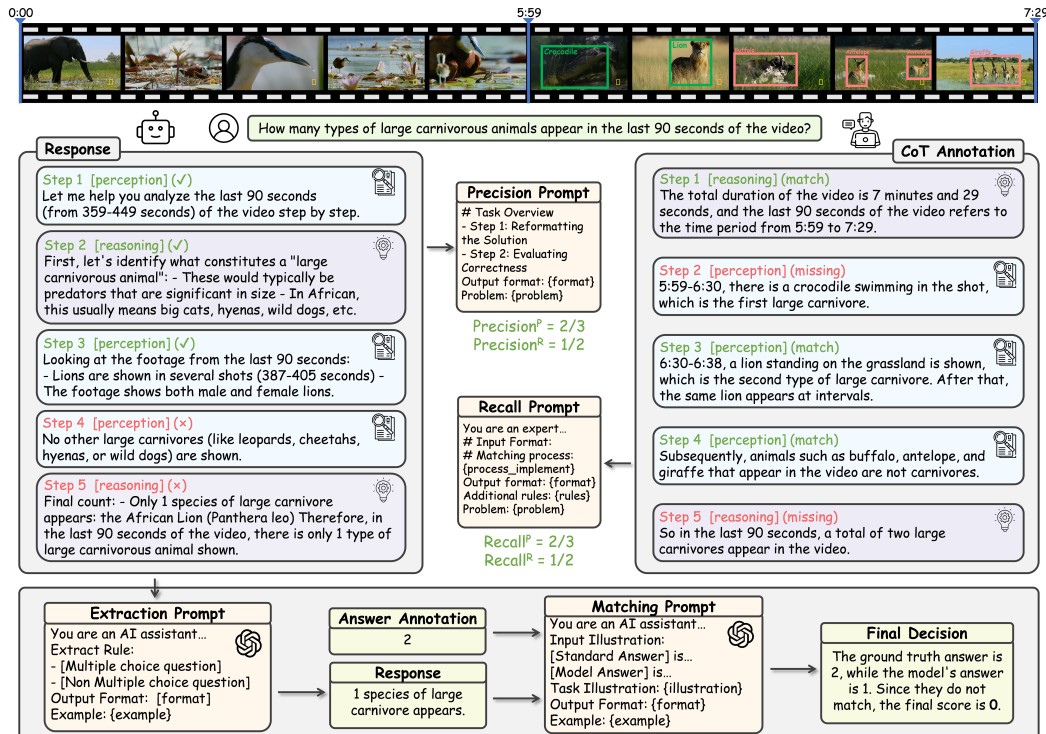

Figure 3: **Overview of VCR-Bench.** For each sample, we provide detailed CoT annotations. During evaluation, we decompose model responses into reasoning steps and match them with reference CoT to compute recall/precision. Final answers are extracted and compared against ground-truth.

### 2.2.2 CoT Quality Evaluation

As described in Section 2.1.2, the question-answer pairs in the VCR-Bench provide accurate and concise reference reasoning steps $\mathcal{R}$. The core of evaluating the model's reasoning content is to establish a matching relationship between the model's reasoning steps $\mathcal{S}$ and the reference reasoning steps $\mathcal{R}$, to determine the correctness of the model's reasoning. To this end, we use GPT4o (OpenAI, 2024b) to decompose the model's reasoning content into $K$ independent and structurally similar sub-steps, and categorize them into two sub-processes, as shown in Eq. 1.

$$\mathcal{S} = \mathcal{S}_p \cup \mathcal{S}_r = \{s_1, s_2, s_3, \dots, s_K\} \tag{1}$$

Then, we evaluate the reasoning process of the model under test based on the following metrics:

**Recall.** For each sub-step $r_i$ in $\mathcal{R}$, we prompt GPT4o to evaluate whether the corresponding content of $r_i$ also appears in $\mathcal{S}$. If the same content appears in $\mathcal{S}$ and is entirely correct — including accurate temporal localization, correct entity recognition, and consistent logical reasoning — then $r_i$ is considered matched and denoted as $r_i^{\text{match}}$. The set of all matched sub-steps is denoted as $\mathcal{R}^{\text{match}}$, and $\mathcal{R}^{\text{match}} = \mathcal{R}_p^{\text{match}} \cup \mathcal{R}_r^{\text{match}}$. The $Recall$ can be calculated as shown in the following Eq. 2.

$$Recall_p = \frac{\left|\mathcal{R}_p^{\text{match}}\right|}{|\mathcal{R}_p|}, Recall_r = \frac{\left|\mathcal{R}_r^{\text{match}}\right|}{|\mathcal{R}_r|}, Recall = \frac{\left|\mathcal{R}^{\text{match}}\right|}{|\mathcal{R}|} \tag{2}$$

The $Recall$ metric comprehensively evaluates the reasoning process by comparing the model's output with the reference solution's key reasoning steps. This metric not only verifies answer correctness but also rigorously examines the logical robustness of the reasoning, effectively eliminating random guessing scenarios, thereby enabling in-depth assessment of the model's reasoning capabilities.

**Precision.** For each sub-step $s_j$ in $\mathcal{S}$, we prompt GPT4o to evaluate based on the content of $\mathcal{R}$ whether $s_j$ is accurate. If $s_j$ matches and is correct according to the content in $\mathcal{R}$, it is considered a correct step, denoted as $s_j^{\text{correct}}$. If $s_j$ does not match or contradicts the content in $\mathcal{R}$, such as errors in the temporal localization of key events, or mistakes in causal reasoning, it is considered an incorrect step, denoted as $s_j^{\text{incorrect}}$. If $s_j$ does not appear in $\mathcal{R}$, or it is impossible to determine whether $s_j$ is correct based on the content in $\mathcal{R}$, it is considered an irrelevant reasoning step in solving the problem, denoted as $s_j^{\text{irrelevant}}$. The set of correct steps and incorrect steps are denoted as $\mathcal{S}^{\text{correct}}$ and $\mathcal{S}^{\text{incorrect}}$. Similarly, both $\mathcal{S}^{\text{correct}}$ and $\mathcal{S}^{\text{incorrect}}$ can be further decomposed into the form as shown in 3.

$$\mathcal{S}^{\text{correct}} = \mathcal{S}_p^{\text{correct}} \cup \mathcal{S}_r^{\text{correct}}, \mathcal{S}^{\text{incorrect}} = \mathcal{S}_p^{\text{incorrect}} \cup \mathcal{S}_r^{\text{incorrect}} \tag{3}$$

Accordingly, the $Precision$ can be calculated as shown in the following Eq. 4 and Eq. 5.

$$Precision_p = \frac{\left|\mathcal{S}_p^{\text{correct}}\right|}{\left|\mathcal{S}_p^{\text{correct}} \cup \mathcal{S}_p^{\text{incorrect}}\right|}, Precision_r = \frac{\left|\mathcal{S}_r^{\text{correct}}\right|}{\left|\mathcal{S}_r^{\text{correct}} \cup \mathcal{S}_r^{\text{incorrect}}\right|} \tag{4}$$

$$Precision = \frac{\left|\mathcal{S}^{\text{correct}}\right|}{\left|\mathcal{S}^{\text{correct}} \cup \mathcal{S}^{\text{incorrect}}\right|} \tag{5}$$

The $Precision$ metrics evaluate the model's output reasoning steps, assessing whether each step is truly reliable and closely related to the answer. By combining $Precision$ and $Recall$ metrics, we can calculate the model's output $F_1$ score as shown in Equation 6 to serve as the final CoT score, thereby enabling more reliable and comprehensive evaluation of the model's CoT response quality.

$$F_1 = 2 \cdot \frac{Precision \cdot Recall}{Precision + Recall} \tag{6}$$

### 2.3 Accuracy Evaluation Strategy

For accuracy evaluation of the models' final results, we first used GPT4o (OpenAI, 2024b) to extract the final answer from the CoT steps. For general QA tasks, GPT4o evaluated correctness against human-annotated reference answers. For specialized tasks like VTG and TSG, we computed the Intersection over Union (IoU) between the extracted and reference answers, considering samples correct if IoU exceeded a threshold (0.7 for VTG, 0.5 for TSG).

## 3 Experiments

### 3.1 Experiment Setup

**Evaluation Models.** To thoroughly evaluate the effectiveness of VCR-Bench, we conducted assessments on multiple models. These include mainstream and powerful closed-source models such as Gemini (1.5 Pro, 2.0 Flash, 2.5 Pro) (Team et al., 2024; Pichai et al., 2024; Comanici et al., 2025), GPT-5 (OpenAI, 2025), GPT-4o (OpenAI, 2024b), o1 (OpenAI, 2024a), o3 (OpenAI, 2025) and Claude (3.5, 4.0) (Anthropic, 2024; 2025), as well as commonly used open-source models like InternVL2.5 (8B, 78B) (Chen et al., 2024g;f;e), VideoLLaMA3 (7B) (Zhang et al., 2025), LLaVA-OneVision (7B, 72B) (Li et al., 2024a), mPLUG-Owl3 (7B) (Ye et al., 2024), MiniCPM-o2.6 (7B) (Yao et al., 2024), Llama-3.2-Vision (11B) (AI@Meta, 2024), Qwen3-VL (8B) (Bai et al., 2025), Qwen2.5-VL (7B, 72B) (Bai et al., 2025), LLaVA-Video (7B, 72B) (Zhang et al., 2024b), Aria (25B) (Li et al., 2024b), and InternVideo2.5 (8B) (Wang et al., 2025). This essentially covers all the mainstream LVLMs currently available. We also provide the evaluation results of human testers for reference.

**Implementation Details.** For models supporting direct video input (e.g., Gemini (Team et al., 2024; Pichai et al., 2024)), videos were processed directly. For models without native video support (e.g., GPT-4o (OpenAI, 2024b)), we extracted 64 frames per video with timestamps and used multi-image

Table 2: **CoT Evaluation Results for Different Models in VCR-Bench.** The best results are **bold** and the second-best are underlined. The $F_1$ represents the final CoT score.

| Model | Perception | | | Reasoning | | | Avg | | |
|---|---|---|---|---|---|---|---|---|---|
| | $Rec$ | $Pre$ | $F_1$ | $Rec$ | $Pre$ | $F_1$ | $Rec$ | $Pre$ | $F_1$ |
| Human Performance | 96.2 | 98.6 | 97.4 | 84.3 | 86.7 | 85.5 | 91.3 | 93.4 | 92.3 |
| *Closed-Source Models* | | | | | | | | | |
| Gemini-2.5-Pro | **74.6** | 62.5 | **68.0** | **70.4** | 67.1 | 68.7 | **71.7** | 63.7 | **67.5** |
| Gemini-2.0-Flash | 52.1 | 66.6 | 58.5 | 57.4 | 64.6 | 60.8 | 54.0 | 62.1 | 57.7 |
| Gemini-1.5-Pro | 47.1 | 57.8 | 51.9 | 54.8 | 54.3 | 54.5 | 49.4 | 54.3 | 51.7 |
| o3 | 66.5 | 53.8 | 59.5 | 56.9 | 59.7 | 58.3 | 67.3 | 55.4 | 60.8 |
| o1 | 52.4 | 70.0 | 59.9 | 66.6 | **71.4** | **68.9** | 56.9 | **70.1** | 62.8 |
| GPT-5 | 71.3 | 57.2 | 63.5 | 59.9 | 67.6 | 63.5 | 65.2 | 59.3 | 62.1 |
| GPT-4o | 51.4 | 61.0 | 55.8 | 55.3 | 52.4 | 53.8 | 52.7 | 56.9 | 54.7 |
| Claude 4 Sonnet | 58.1 | 46.0 | 51.3 | 51.0 | 53.0 | 52.0 | 60.1 | 46.0 | 52.1 |
| Claude 3.5 Sonnet | 47.7 | 58.1 | 52.4 | 49.1 | 47.5 | 48.3 | 47.6 | 53.6 | 50.4 |
| *Open-Source Models* | | | | | | | | | |
| InternVL2.5-8B | 16.1 | 52.6 | 24.6 | 33.0 | 36.9 | 34.8 | 22.1 | 38.2 | 28.0 |
| InternVL2.5-78B | 18.7 | 74.1 | 29.9 | 35.2 | 53.9 | 42.6 | 23.9 | 56.8 | 33.7 |
| VideoLLaMA3-7B | 20.2 | 52.2 | 29.1 | 39.1 | 39.9 | 39.5 | 26.6 | 40.1 | 32.0 |
| LLaVA-OneVision-7B | 10.1 | 92.3 | 18.3 | 28.7 | 51.2 | 36.8 | 16.7 | 55.1 | 25.6 |
| LLaVA-OneVision-72B | 14.1 | 94.7 | 24.5 | 35.5 | 58.3 | 44.1 | 20.8 | 61.5 | 31.1 |
| mPLUG-Owl3-7B | 6.0 | 86.5 | 11.1 | 20.7 | 43.7 | 28.1 | 10.4 | 45.4 | 17.0 |
| MiniCPM-o2.6-8B | 27.5 | 49.4 | 35.3 | 34.6 | 35.0 | 34.8 | 29.9 | 38.7 | 33.8 |
| Llama-3.2-11B-Vision | 2.1 | 86.4 | 4.2 | 6.8 | 52.5 | 12.0 | 3.6 | 52.5 | 6.8 |
| Qwen3-VL-8B | 62.5 | 41.1 | 49.6 | 50.4 | 44.6 | 47.3 | 61.0 | 41.8 | 49.6 |
| Qwen2.5-VL-7B | 31.7 | 53.4 | 39.8 | 34.7 | 37.4 | 36.0 | 33.4 | 44.6 | 38.2 |
| Qwen2.5-VL-72B | 46.2 | 60.2 | 52.3 | 47.4 | 46.1 | 46.7 | 47.5 | 53.8 | 50.5 |
| LLaVA-Video-7B | 11.1 | 95.7 | 19.9 | 33.1 | 52.0 | 40.4 | 18.1 | 56.4 | 27.3 |
| LLaVA-Video-72B | 15.6 | 95.3 | 26.9 | 39.8 | 57.1 | 46.9 | 23.2 | 60.6 | 33.6 |
| Aria-25B | 18.5 | 68.6 | 29.1 | 36.2 | 52.3 | 42.8 | 23.9 | 56.0 | 33.5 |
| InternVideo2.5-8B | 6.9 | **98.4** | 12.9 | 26.1 | 61.3 | 36.6 | 12.6 | 66.0 | 21.2 |

input for evaluation. All other parameters followed official specifications. During inference, models answered questions step-by-step using our CoT prompt: "*Please provide a step-by-step solution to the given question.*" All other prompts used during evaluation are provided in the Appendix E.

## 3.2 CoT Evaluation Results

We first evaluated the output CoT steps of each model, all prompts used during evaluation are provided in the Appendix E, and the experimental results are shown in Table 2. From the results, it can be observed that the quality of output CoT varies significantly across different models, and the overall CoT scores are not particularly high. Among them, the Gemini 2.5 Pro (Comanici et al., 2025) model, known for its strong reasoning capabilities, achieved a comprehensive CoT score of 67.5, ranking first among all models. Further analysis of the results leads us to the following conclusions:

**Closed-source models and large-scale parameter models possess stronger reasoning capabilities.** As shown in the results of Table 2, the CoT evaluation CoT scores of common closed-source models are generally higher than those of open-source models. Additionally, for the same open-source model with different parameter sizes, such as Qwen2.5-VL 7B and 72B (Bai et al., 2025), the model with larger parameters achieves a higher CoT score. This reflects that video CoT reasoning places high demands on the overall performance of LVLMs, and only models with larger parameters can ensure better step-by-step analysis and reasoning capabilities.

**A more common issue that models encounter during multi-step reasoning is omission rather than inaccuracy.** Experimental results show that most models achieve higher precision than recall. Models with weaker CoT ability (*e.g.*, LLaVA-Video-7B (Zhang et al., 2024b)) often generate

Table 3: **Accuracy Evaluation Results for Different Models in VCR-Bench.** The best results are **bold** and the second-best are underlined.

| Model | FTR | VTC | VTG | VKR | TSR | VPA | TSG | Avg |
|---|---|---|---|---|---|---|---|---|
| Human Performance | 90.2 | 95.6 | 90.0 | 60.2 | 95.2 | 96.3 | 80.3 | 86.8 |
| *Closed-Source Models* | | | | | | | | |
| Gemini-2.5-Pro | 56.1 | 56.4 | **62.7** | **75.9** | 56.5 | **62.2** | 0.0 | 54.6 |
| Gemini-2.0-Flash | 66.2 | 51.2 | 62.0 | 64.4 | 54.1 | 58.1 | 4.2 | 51.7 |
| Gemini-1.5-Pro | 55.1 | 45.3 | 52.9 | 62.0 | 45.0 | 45.6 | 0.7 | 44.0 |
| o3 | 52.8 | 45.3 | 49.0 | 61.8 | 44.4 | 53.2 | 4.9 | 44.8 |
| o1 | **66.7** | 52.2 | 56.9 | 74.3 | 61.0 | 60.2 | 0.0 | **56.7** |
| GPT-5 | 54.7 | **63.7** | 53.7 | 56.5 | **65.9** | 54.9 | **15.9** | 51.4 |
| GPT-4o | 54.7 | 49.1 | 44.8 | 68.6 | 48.9 | 57.6 | 2.8 | 46.9 |
| Claude 4 Sonnet | 50.9 | 46.6 | 48.3 | 59.9 | 47.4 | 48.9 | 6.3 | 44.3 |
| Claude 3.5 Sonnet | 45.3 | 46.3 | 34.3 | 64.2 | 44.0 | 49.3 | 0.7 | 41.0 |
| *Open-Source Models* | | | | | | | | |
| InternVL2.5-8B | 32.7 | 29.8 | 11.9 | 33.3 | 25.9 | 30.9 | 0.7 | 23.9 |
| InternVL2.5-78B | 40.9 | 39.8 | 9.8 | 52.9 | 29.6 | 39.6 | 0.0 | 30.9 |
| VideoLLaMA3-7B | 44.7 | 36.6 | 24.5 | 43.1 | 36.3 | 39.6 | 0.7 | 32.5 |
| LLaVA-OneVision-7B | 35.8 | 34.8 | 24.5 | 39.9 | 37.8 | 41.0 | 0.0 | 30.7 |
| LLaVA-OneVision-72B | 47.8 | 42.2 | 25.9 | 52.3 | 45.9 | 38.1 | 0.0 | 36.4 |
| mPLUG-Owl3-7B | 13.2 | 6.2 | 2.8 | 5.9 | 15.6 | 7.2 | 0.0 | 7.3 |
| MiniCPM-o2.6-8B | 31.4 | 30.4 | 12.6 | 43.8 | 30.4 | 38.1 | 0.0 | 26.9 |
| Llama-3.2-11B-Vision | 4.4 | 4.3 | 7.0 | 6.5 | 6.7 | 5.8 | 0.0 | 4.9 |
| Qwen3-VL-8B | 37.4 | 37.8 | 47.6 | 54.2 | 29.5 | 32.6 | 3.5 | 35.0 |
| Qwen2.5-VL-7B | 37.1 | 26.7 | 29.4 | 47.1 | 34.8 | 36.0 | 0.7 | 30.4 |
| Qwen2.5-VL-72B | 45.0 | 39.9 | 34.1 | 56.2 | 38.1 | 48.9 | 2.1 | 37.9 |
| LLaVA-Video-7B | 47.2 | 36.6 | 18.9 | 41.8 | 40.7 | 40.3 | 0.0 | 32.5 |
| LLaVA-Video-72B | 49.7 | 49.1 | 17.5 | 49.7 | 43.7 | 43.2 | 0.0 | 36.6 |
| Aria-25B | 45.3 | 45.0 | 33.6 | 56.2 | 43.7 | 38.8 | 2.8 | 38.2 |
| InternVideo2.5-8B | 40.9 | 43.5 | 14.0 | 41.2 | 48.1 | 41.7 | 0.0 | 33.0 |

only one or two steps, further widening the gap. This suggests that while most generated steps are accurate, many critical reasoning steps are still omitted.

**The logical reasoning performance of the models is generally stronger than their visual perception performance.** Models generally perform better in logical reasoning than visual perception. Quantitative analysis shows their average reasoning score (mean CoT 45.8) exceeds perception (mean CoT 38.7), with open-source models showing larger gaps. This indicates that LVLMs' main bottleneck in complex video reasoning lies in visual perception and comprehension.

### 3.3 ACCURACY EVALUATION RESULTS

As shown in Table 3, we evaluated the final answer accuracy of all models across different dimensions. Combined with the results from Table 2, we can draw the following conclusions:

**The CoT evaluation results are highly positively correlated with the final answer evaluation results.** As shown in Figure 4, experimental results demonstrate a strong positive correlation (r=0.90) between models' CoT reasoning quality and final answer accuracy. This robust relationship confirms that effective CoT reasoning is critical for successful video question answering, with higher-quality CoT steps consistently leading to more accurate final responses.

**Models with stronger instruction-following capabilities can achieve relatively higher CoT scores.** A closer examination of Figure 4 reveals that some models exhibit relatively high accuracy but low CoT scores, such as LLaVA-Video-7B (Zhang et al., 2024b) and LLaVA-OneVision-7B (Li et al., 2024a). These models generally struggle to properly follow CoT instructions—even when provided with CoT prompts, their outputs remain overly concise, and their reasoning processes are insufficiently detailed, resulting in lower CoT scores. However, models like Qwen2.5-VL (Bai et al.,

Table 4: Accuracy Evaluation Results for Different Durations.

| Model | Short | Med | Long | Avg |
|---|---|---|---|---|
| *Closed-Source Models* | | | | |
| Gemini-2.5-Pro | **54.2** | 58.4 | 47.2 | 54.6 |
| Gemini-2.0-Flash | 44.2 | 60.3 | 53.5 | 51.7 |
| Gemini-1.5-Pro | 37.4 | 49.9 | 48.7 | 44.0 |
| o3 | 41.5 | 47.1 | 48.4 | 44.8 |
| o1 | 53.6 | **61.3** | **54.7** | **56.7** |
| GPT-5 | 51.4 | 54.2 | 45.0 | 51.4 |
| GPT-4o | 44.4 | 48.7 | 49.7 | 46.9 |
| Claude 4 Sonnet | 42.4 | 46.8 | 44.1 | 44.3 |
| Claude 3.5 Sonnet | 39.8 | 42.2 | 41.4 | 41.0 |
| *Open-Source Models* | | | | |
| InternVL2.5-8B | 20.7 | 25.7 | 28.3 | 23.9 |
| InternVL2.5-78B | 30.4 | 30.5 | 32.6 | 30.9 |
| VideoLLaMA3-7B | 30.2 | 38.2 | 26.7 | 32.5 |
| LLaVA-OneVision-7B | 29.2 | 33.4 | 28.9 | 30.7 |
| LLaVA-OneVision-72B | 35.1 | 40.6 | 31.0 | 36.4 |
| mPLUG-Owl3-7B | 6.1 | 9.9 | 4.8 | 7.3 |
| MiniCPM-o2.6-8B | 27.5 | 26.0 | 26.7 | 26.9 |
| Llama-3.2-11B-Vision | 5.3 | 5.1 | 3.7 | 4.9 |
| Qwen3-VL-8B | 34.0 | 37.9 | 30.9 | 35.0 |
| Qwen2.5-VL-7B | 27.1 | 34.0 | 31.6 | 30.4 |
| Qwen2.5-VL-72B | 33.4 | 42.8 | 39.8 | 37.9 |
| LLaVA-Video-7B | 31.7 | 33.4 | 32.6 | 32.5 |
| LLaVA-Video-72B | 35.5 | 40.6 | 38.5 | 37.9 |
| Aria-25B | 36.4 | 39.9 | 39.6 | 38.2 |
| InternVideo2.5-8B | 31.5 | 35.0 | 32.6 | 33.0 |

Table 5: Accuracy Evaluation Results under Different Settings.

| Model | Text | 1 Frame | Direct | CoT |
|---|---|---|---|---|
| *Closed-Source Models* | | | | |
| Gemini-2.0-Flash | **13.8** | **25.2** | 44.8 | **51.7** |
| GPT-4o | 9.8 | 21.6 | **46.3** | 46.9 |
| Claude 3.5 Sonnet | 9.1 | 11.3 | 39.6 | 41.0 |
| *Open-Source Models* | | | | |
| InternVL2.5-78B | 7.2 | 18.7 | 35.4 | 30.9 |
| Qwen2.5-VL-72B | 12.7 | 16.7 | 42.7 | 37.9 |

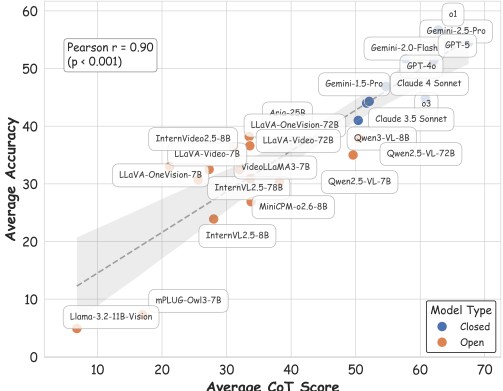

Figure 4: Correlation between CoT Evaluation Results and Accuracy Evaluation Results.

2025), which demonstrate stronger instruction-following capabilities, produce more comprehensive reasoning chains, thus achieving comparatively higher CoT scores.

**The spatiotemporal grounding capabilities of the models are generally weak.** The TSG task proves exceptionally challenging, with even the top model (GPT-5 (OpenAI, 2025)) achieving merely 15.9% accuracy, while many models fail completely. This stems from the task's unique demands: (1) combined spatiotemporal reasoning (temporal localization + coordinate output), and (2) current models' fundamental limitations in extracting precise spatial coordinates from video data. For concrete examples, please refer to Figure 6 in the Appendix F.

## 3.4 MORE EVALUATION RESULTS

**Accuracy Evaluation Results for Different Durations.** We also statistically analyzed the model's performance across videos of different durations, as shown in Table 4. The results indicate that the model generally achieves better performance on medium-length videos. In comparison, long videos contain more complex temporal information and richer content, which poses greater challenges for the model's comprehension. As for short videos, since our dataset is primarily based on manual annotations and corrections, human annotators tend to find them easier to understand and are thus able to produce more in-depth and sophisticated annotations. Meanwhile, the model shows significant deficiencies in the TSG dimension, which mainly consists of short videos. This partially contributes to its weaker performance on short-form content.

**Accuracy Evaluation Results under Different Settings.** To further validate the rationality of VCR-Bench, we conducted experiments under different settings, including: text-only input without video, text plus a single frame extracted from video, and full text plus video with direct answering (without CoT), compared with our standard setup of full text plus video with CoT answering. As shown in Table 5, both the text-only and single-frame input settings lead to significant performance degradation, indicating that our question-answer data highly depend on video content and temporal information. Meanwhile, for stronger closed-source models, using CoT prompting results in higher accuracy than

direct answering, whereas the opposite is true for weaker open-source models. This demonstrates that effective CoT reasoning heavily relies on the model's overall capability—only models with sufficiently strong reasoning skills can fully benefit from CoT.

## 4 CONCLUSION

We introduce VCR-Bench, the first benchmark tailored to evaluate the Chain-of-Thought (CoT) reasoning capabilities of LVLMs in video understanding tasks. It consists of a high-quality dataset of 859 diverse videos and 1,034 QA pairs across seven task types, each annotated with detailed CoT reasoning references. We propose a novel evaluation framework that assesses reasoning quality through recall, precision, and the $F_1$ score. Comprehensive evaluations show that current LVLMs, including top-performing models, have significant limitations, with the best model achieving only a 67.5 CoT score and most open-source models scoring below 40. This highlights the need for substantial improvement in video-grounded reasoning. Further experimental results also indicate that the primary capability bottleneck for most current models in handling complex video questions lies in the visual perception step. VCR-Bench provides a standardized framework to drive research progress in this critical area.

## 5 ETHICS STATEMENT

This work adheres to ethical research standards in data collection, model training, and evaluation. All datasets used in this study are publicly available research datasets, collected and released under their respective licenses. No private or personally identifiable information (PII) was used.

## 6 REPRODUCIBILITY STATEMENT

We are committed to ensuring the reproducibility of our results. All prompts used during evaluation are provided in the appendices. Additionally, all datasets and evaluation scripts will be made publicly available.

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

APPENDIX

# A RELATED WORK

## A.1 LVLMs FOR VIDEO UNDERSTANDING

The rapid advancement of image-based LVLMs (Chen et al., 2024c; Lin et al., 2023; Ye et al., 2024; Maaz et al., 2023) has significantly boosted video understanding and question answering capabilities, revitalizing AI research. Early attempts like VideoChat and Video-ChatGPT (Maaz et al., 2023) paved the way for recent advancements such as CogVLM2-Video (Hong et al., 2024), InternVL2 (Chen et al., 2024g;f), and LLaVA-Video (Zhang et al., 2024b), which process videos as image sequences by leveraging powerful image comprehension. To address the computational challenges of high frame rates and long videos, techniques like QFormer-based feature extraction in InternVideo2 (Wang et al., 2024b) and Video-LLaMA (Zhang et al., 2023), and adaptive pooling in PLLaVA (Xu et al., 2024) have been developed. With the enhancement of model capabilities and the increasing complexity of tasks, the strong reasoning and thinking abilities of LVLMs in the field of video understanding are receiving growing attention.

## A.2 VIDEO UNDERSTANDING BENCHMARKS

Traditional video understanding benchmarks focus on evaluating specific model capabilities in particular scenarios. For example, MSRVTT-QA (Xu et al., 2017), ActivityNet-QA (Yu et al., 2019), and NExT-QA (Xiao et al., 2021) test basic action recognition and video question answering, while MMBench (Xu et al., 2023), SEED-Bench (Li et al., 2023), and MVBench (Li et al., 2024c) assess short video clips. Benchmarks like LongVideoBench (Wu et al., 2024), Video-MME (Fu et al., 2024), and LVBench (Wang et al., 2024a) provide longer videos and more diverse tasks. Latest work, such as V2P-Bench (Zhao et al., 2025b), has constructed a set of data based on visual prompts by simulating human-computer interactions. However, these tasks are generally simple and do not require complex reasoning from models. Recently, there has been growing interest in video CoT reasoning tasks. VideoEspresso (Han et al., 2024) uses keyframe captions for complex scene reasoning, MMVU (Zhao et al., 2025a) introduces annotated educational video reasoning questions, and VideoMMMU (Hu et al., 2025) focuses on knowledge reasoning from subject explanation videos. While these efforts aim to measure video CoT reasoning, their scenarios are limited, and they primarily evaluate final results rather than the reasoning process itself.

## A.3 REASONING EVALUATION

In the multimodal domain, research on reasoning process evaluation is limited and mostly focused on images. Early works, such as MathVista (Lu et al., 2023), MathVerse (Zhang et al., 2024a), and OlympiadBench (He et al., 2024), targeted narrow scientific tasks. More recent benchmarks like M³CoT (Chen et al., 2024d) and SciVerse (Guo et al., 2024) extend to general image reasoning but still lack comprehensive process evaluation. LlamaV-o1 (Thawakar et al., 2025) introduces a multi-dimensional framework, and MME-CoT (Jiang et al., 2025) evaluates image reasoning by aligning model steps with annotations and computing $F_1$ scores. These methods can be adapted for video reasoning.

# B MORE EXPERIMENTAL RESULTS

**Performance Analysis of Video Reasoning Models.** Recent work has also introduced several RL-based approaches specifically designed for video reasoning. We selected two representative models, Video-R1-7B (Feng et al., 2025) and LongVILA-R1-7B Chen et al. (2025), and analyzed their performance on VCR-Bench. The results are presented in Table 6 and Table 7 below. Video-R1-7B (Feng et al., 2025) is built upon Qwen2.5-VL-7B (Bai et al., 2025) as the base model and further trained via reinforcement learning. Experimental results show that, after targeted video reasoning training, Video-R1-7B achieves superior performance on VCR-Bench, particularly in reasoning steps, compared to its base model.

Table 6: CoT Evaluation Results for Video Reasoning Models in VCR-Bench.

| Model | Perception | | | Reasoning | | | Avg | | |
|---|---|---|---|---|---|---|---|---|---|
| | $Rec$ | $Pre$ | $F_1$ | $Rec$ | $Pre$ | $F_1$ | $Rec$ | $Pre$ | $F_1$ |
| Qwen2.5-VL-7B | 31.7 | 53.4 | 39.8 | 34.7 | 37.4 | 36.0 | 33.4 | 44.6 | 38.2 |
| Video-R1-7B | 35.0 | 20.3 | 25.3 | 50.4 | 59.7 | 54.7 | 38.6 | 41.3 | 39.9 |
| LongVILA-R1-7B | 39.8 | 18.6 | 25.4 | 50.7 | 28.4 | 36.4 | 39.9 | 20.6 | 27.1 |

Table 7: Accuracy Evaluation Results for Video Reasoning Models in VCR-Bench.

| Model | FTR | VTC | VTG | VKR | TSR | VPA | TSG | Avg |
|---|---|---|---|---|---|---|---|---|
| Qwen2.5-VL-7B | 37.1 | 26.7 | 29.4 | 47.1 | 34.8 | 36.0 | 0.7 | 30.4 |
| Video-R1-7B | 32.8 | 34.6 | 22.9 | 47.9 | 46.7 | 24.2 | 1.8 | 33.6 |
| LongVILA-R1-7B | 33.3 | 36.0 | 17.5 | 41.2 | 28.9 | 30.2 | 0.7 | 27.2 |

**Comparison of model performance across different judges.** In our VCR-Bench evaluation, we use GPT-4o (OpenAI, 2024b) as the judge model. To further validate the robustness of our assessment, we additionally select several other strong reasoning models (Comanici et al., 2025; Guo et al., 2025b; OpenAI, 2025) as judges to evaluate the model-generated reasoning outputs. As shown in Table 8, the results indicate that different judge models lead to variations in the final evaluation scores. However, the overall ranking remains largely consistent, demonstrating the relative reliability of using a single judge model for evaluation.

Table 8: Comparison of performance using different models as judges.

| Test Model | Judge Model | | | |
|---|---|---|---|---|
| | GPT-4o | Gemini-2.5 | o3 | doubao-1.5 |
| Gemini-2.0-Flash | 51.7 | 52.7 | 53.6 | 51.3 |
| GPT-4o | 46.9 | 49.9 | 51.1 | 42.3 |
| Qwen2.5-VL-7B | 30.4 | 32.0 | 32.4 | 32.2 |
| Qwen2.5-VL-72B | 37.9 | 39.7 | 41.2 | 39.1 |

**The impact of different input frame counts on model performance.** During our evaluation, we selected 64 frames as the input setting for most models. Here, we conduct a frame-number ablation study based on Qwen2.5-VL-7B (Bai et al., 2025), where inputs with varying numbers of frames are sampled uniformly, and the resulting model performance is compared, as shown in Table 9. The results indicate that with fewer frames, model performance drops significantly due to insufficient information. Conversely, when the number of frames exceeds 64, the performance gains start to diminish. Moreover, excessive frame inputs not only increase inference costs but also exceed the input capacity of some models. Therefore, we ultimately chose 64 frames as our standard input.

# C MORE DISCUSSION

**Different Reasoning Traces.** When dealing with complex reasoning problems, multiple valid reasoning paths may exist, which could introduce certain biases into the evaluation. We have carefully considered this issue during the data annotation process and took measures to minimize the impact of divergent reasoning routes. First, most samples in our evaluation set were intentionally designed to have a fixed reasoning trajectory or to include unavoidable key clues that guide the reasoning process. Compared with image data, video inherently provides a clear and unidirectional temporal structure, making it easier to identify crucial moments and constrain the possible reasoning paths. This property naturally reduces ambiguity arising from multiple potential reasoning routes. In addition, we filtered out samples with unclear or poorly defined reasoning processes to further ensure the reliability and consistency of the evaluation.

Table 9: Effect of different input frame counts on model performance.

| Frames | 16 | 32 | 64 | 128 |
|---|---|---|---|---|
| Accuracy (%) | 13.5 | 33.7 | 36.9 | 37.2 |

Table 10: Probability (%) of each model generating a fully correct CoT reasoning process conditioned on producing the correct final answer.

| Model | Probability | Model | Probability |
|---|---|---|---|
| InternVideo2.5-8B | 7.9 | LLaVA-OneVision-72B | 17.6 |
| InternVL2.5-8B | 22.2 | LLaVA-OneVision-7B | 11.0 |
| Gemini-1.5-Pro | 44.2 | Gemini-2.0-Flash | 56.6 |
| VideoLLaMA3-7B | 21.2 | GPT-4o | 49.9 |
| o1 | 58.0 | Claude 3.5 Sonnet | 52.4 |
| mPLUG-Owl3-7B | 5.3 | InternVL2.5-78B | 24.4 |
| MiniCPM-o2.6-8B | 28.2 | Qwen2.5-VL-7B | 40.1 |
| Llama-3.2-11B-Vision | 2.0 | Qwen2.5-VL-72B | 53.4 |
| Aria-25B | 12.7 | LLaVA-Video-72B | 18.4 |
| LLaVA-Video-7B | 11.3 | - | - |

**Low Performance on TSG Task.** Among the seven tasks included in our benchmark, the TSG task is particularly challenging, as nearly all evaluated models perform poorly on this task. We attribute this phenomenon to several potential reasons. First, the TSG task remains relatively new in multimodal video reasoning, and current models may lack sufficient training data tailored to this setting. Second, spatial perception itself is inherently difficult for multimodal models. Many existing models struggle with spatial localization, as illustrated in the case studies in Appendix Figure 6. In our TSG formulation, the model is required not only to perform localization but also to conduct additional reasoning to justify the localization process, which substantially increases the task difficulty. Nevertheless, we argue that this task setting provides a rigorous assessment of a model's spatiotemporal perception capabilities and offers valuable insights for future model development. As multimodal models continue to advance, the importance and necessity of the TSG task will become increasingly evident.

**Further analysis of the relationship between CoT score and accuracy.** In Section 3.3, we demonstrated the correlation between the CoT score and accuracy, confirming a strong relationship between the two metrics. Here, we further provide statistics on the probability that each model achieves both a correct accuracy prediction and a full CoT score, as shown in Table 10. As the results indicate, consistent with the patterns illustrated in Figure 1 and Figure 7, models may still exhibit issues in their reasoning traces even when their final accuracy is correct. This further underscores the necessity of incorporating CoT-based evaluation.

**Annotation Details.** Our dataset was constructed using a combination of model-generated content and human annotation. During the human annotation phase, we implemented several measures to minimize labeling errors: all annotators are experienced practitioners in the VLM community, with solid expertise in multimodal video understanding and reasoning tasks; we conducted unified training to ensure consistent annotation standards; annotators were instructed to strictly follow the objective content presented in the videos and to avoid introducing any subjective inference or judgment; a cross-checking mechanism was employed, in which each annotation was reviewed by two to three additional annotators; finally, to guarantee high-quality annotations, any samples that could introduce ambiguity or disagreement were excluded.

**Comparison with Existing Long Video Reasoning Benchmarks.** Recent work has also begun exploring video reasoning benchmarks, such as LongVideo-Reason-eval (Chen et al., 2025). These studies evaluate models from several capability dimensions and design corresponding question types for each dimension. Although their evaluation sets include reasoning processes to some extent, the assessment essentially remains result-oriented. In contrast, our VCR-Bench fundamentally differs from these approaches. While also providing intermediate reasoning steps, its core contribution lies

in introducing a step-by-step scoring mechanism for the reasoning process itself. This enables a more comprehensive and fine-grained reflection of a model's true reasoning capability.

## D LIMITATION AND BROADER IMPACT

**Limitation.** Our evaluation methodology relies heavily on the assessment provided by GPT-4o. Limited by the capabilities of GPT-4o itself, it cannot achieve true and complete objectivity, which is a limitation of our evaluation system. We are also exploring ways to minimize the use of model-based evaluation methods to make the evaluation system more objective and fair.

**Broader Impact.** The VCR-Bench constructs an innovative method for cot evaluation of video question answering in the multimodal domain, which will be highly beneficial for more comprehensively validating the reasoning capabilities of LVLMs, enhancing the performance of LVLMs in the field of video understanding, and providing more possibilities for the practical applications of large models (such as VR headsets and mobile assistants).

## E PROMPT TEMPLATE

---

**Recall Evaluation Prompt**

You are an expert system for verifying solutions to video-based problems. Your task is to match the ground truth middle steps with the provided solution.

**INPUT FORMAT:**
1. Problem: The original question/task
2. A Solution of a model
3. Ground Truth: Essential steps required for a correct answer

**MATCHING PROCESS:**
You need to match each ground truth middle step with the solution:

**Match Criteria:**
- The middle step should exactly match in the content or is directly entailed by a certain content in the solution
- All the details must be matched, including the specific value and content
- You should judge all the middle steps for whether there is a match in the solution

**Step Types:**
1. Logical Inference Steps
- Contains exactly one logical deduction
- Must produce a new derived conclusion
- Cannot be just a summary or observation

2. Video Description Steps
- Pure visual observations
- Only includes directly visible elements
- No inferences or assumptions
- Contains event time

**OUTPUT FORMAT:**
JSON array of judgments:

```
[
  {
    "step": ground truth middle step,
    "step_type": "Video Description Steps|
    Logical Inference Steps",
```

---

```
        "judgment": "Matched" | "Unmatched"
    }
]
```
**ADDITIONAL RULES:**
1. Only output the json array with no additional information.
2. Judge each ground truth middle step in order without omitting any step.

Here is the problem, answer, solution, and the ground truth middle steps:
**[Problem]**: {question}
**[Answer]**: {answer}
**[Solution]**: {solution}

---

Precision Evaluation Prompt

Given a solution with multiple reasoning steps for a video-based problem, reformat it into well-structured steps and evaluate their correctness.

**Step 1: Reformatting the Solution**
Convert the unstructured solution into distinct reasoning steps while:
- Preserving all original content and order
- Not adding new interpretations
- Not omitting any steps

**Step Types**
1. Logical Inference Steps
- Contains exactly one logical deduction
- Must produce a new derived conclusion
- Cannot be just a summary or observation

2. Video Description Steps
- Pure visual observations
- Only includes directly visible elements
- No inferences or assumptions
- Contains event time

3. Background Review Steps:
- Repetition or review of the problem
- Not directly related to solving the problem.

**Step Requirements**
- Each step must be atomic (one conclusion per step)
- No content duplication across steps
- Initial analysis counts as background information
- Final answer determination counts as logical inference

**Step 2: Evaluating Correctness**
Evaluate each step against:
**Ground Truth Matching**
For video descriptions:
- Key elements must match ground truth descriptions
For logical inferences:
- Conclusion must EXACTLY match or be DIRECTLY entailed by ground truth
For Background review:
- Without special circumstances are deemed to be redundant

**Reasonableness Check (if no direct match)**

If Step:
- Premises must not contradict any ground truth or correct answer
- Logic is valid
- Conclusion must not contradict any ground truth
- Conclusion must support or be neutral to correct answer
- Helpful in solving the problem, non-redundant steps
this Step be viewed as matched.

**Judgement Categories**
- "Match": Aligns with ground truth
- "Wrong": Contradictory with ground truth
- "Redundant": Redundant steps that do not help solve the problem

**Output Requirements**
1. The output format MUST be in valid JSON format without ANY other content.
2. For highly repetitive patterns, output it as a single step.
3. Output maximum 35 steps. Always include the final step that contains the answer.

**Output Format**

```
[
  {
    "step": "reformatted the solution step",
    "step_type": "Video Description Steps|
                  Logical Inference Steps|
                  Background Review Steps",
    "reasons_for_judgment": "The reason for judging...",
    "judgment": "Matched|Wrong|Redundant"
  }
]
```

**Input Data**
**[Problem]:** {question}
**[Solution]:** {solution}
**[Ground Truth Information]:** {gt_annotation}

---

Answer Extraction Prompt

You are an AI assistant who will help me to extract an answer of a question. You are provided with a question and a response, and you need to find the final answer of the question.

**Extract Rule:**
[Multiple choice question]
1. The answer could be answering the option letter or the value. You should directly output the choice letter of the answer.
2. You should output a single uppercase character in A, B, C, D, E, F, G, H, I (if they are valid options), and Z.
3. If the answer is about a certain time period, such as from 1 minute 30 seconds to 2 minutes 30 seconds, it should be given in the format [90, 150].
4. If the meaning of all options are significantly different from the final answer, output Z.

[Non Multiple choice question]
1. Output the final value of the answer. It could be hidden inside the last step of calculation or inference. Pay attention to what the question is asking for to extract the value of the answer.
2. The final answer could also be a short phrase or sentence.

3. If the response doesn't give a final answer, output Z.

**Output Format:**
Directly output the extracted answer of the response

**Example 1:**
Question: What is the main object in image?
Options: A. teddy bear B. rabbit C. cat D. dog
Response: a cute teddy bear
Your output: A

**Example 2:**
Question: What is the main object in image?
Options: A. teddy bear B. rabbit C. cat D. dog
Answer: Spider
Your output: Z

**Example 3:**
Question:
On a flat playground, choose a Cartesian $Oxy$ coordinate system (in units of meters). A child running at a constant velocity $V = 1\,\mathrm{m/s}$ around a heart-shaped path satisfies the following order-6 algebraic equation:

$$\left(x^2 + y^2 - L^2\right)^3 - Lx^2y^3 = 0, \quad L = 10.$$

Response:
Given the radius of curvature $r \approx 10\,\mathrm{m}$:
Using the radius of curvature:

$$a_c = \frac{V^2}{r} = \frac{(1\,\mathrm{m\,s^{-1}})^2}{10\,\mathrm{m}} = 0.1\,\mathrm{m\,s^{-2}}$$

The magnitude of the acceleration is the sum of tangential and centripetal components. Since tangential acceleration is zero (constant speed):

$$\text{Magnitude of acceleration} = a_c = \boxed{0.1\,\mathrm{m\,s^{-2}}}$$

Your output: $0.1\,\mathrm{m\,s^{-2}}$

**Example 4:** Question: Given the sequence of images, please identify the image consistent with the text description: Clean And Jerk.
A. ¡image1¿
B. ¡image2¿
C. ¡image3¿
D. ¡image4¿,
Response: Based on the temporal analysis, Answer: The second image
Your output: B

**Example 5:**
Question: Please specify the start and end time of the event 'An intro screen appears with a gray background and the red and white words that say PRO TRAINING.'
Response: The start and end time of the event 'An intro screen appears with a gray background and the red and white words that say PRO TRAINING.' are as follows:
Step 1: Identify the start time of the event.
The video starts at 00:00. The intro screen is visible from the very beginning.
Start time: 00:00
Step 2: Identify the end time of the event.
The intro screen starts to disappear and transition to the next scene at 00:11. By 00:12, the intro screen is completely gone. So, the intro screen is visible until 00:11. End time: 00:11

Therefore, the event 'An intro screen appears with a gray background and the red and white words that say PRO TRAINING' starts at 00:00 and ends at 00:11.
Your output: [0,11]

**Example 6:**
Question: In the diagram, $OA = 15, OP = 9$ and $PB = 4$. Determine the equation of the line through $A$ and $B$. Explain how you got your answer.
Response: The equation of the line through $A$ and $B$ is $y = -\frac{1}{3}x + 12$.
To find the equation of the line through $A$ and $B$, we first need to find the slope of the line.
The slope of a line passing through two points $(x_1, y_1)$ and $(x_2, y_2)$ is given by the formula:

$$\text{Slope} = \frac{y_2 - y_1}{x_2 - x_1}$$

Using the coordinates of points $A(0, 15)$ and $B(4, 0)$, we can calculate the slope as follows:

$$\text{Slope} = \frac{0 - 15}{4 - 0} = \frac{-15}{4} = -\frac{15}{4}$$

Now that we have the slope, we can use the point-slope form of a linear equation to find the equation of the line. The point-slope form is given by:

$$y - y_1 = m(x - x_1)$$

where $(x_1, y_1)$ is a point on the line and $m$ is the slope. In this case, we can use point $A(0, 15)$ and the slope $-\frac{15}{4}$:

$$y - 15 = -\frac{15}{4}(x - 0)$$

Simplifying the equation, we get:

$$y - 15 = -\frac{15}{4}x$$

$$y = -\frac{15}{4}x + 15$$

Therefore, the equation of the line through $A$ and $B$ is $y = -\frac{15}{4}x + 15$.
Your output: $y = -\frac{15}{4}x + 15$

**Example 7:**
Question: ¡image¿ ¡image¿ ¡image¿ ¡image¿ ¡image¿ ¡image¿ ¡image¿ ¡image¿ ¡image¿In the paper 'An Object is Worth 64x64 Pixels: Generating 3D Object via Image Diffusion', what aspect of the proposed method does this figure aim to demonstrate?
Response: the potential of this 3D generation paradigm to generate objects with PBR materials
Your output: the potential of this 3D generation paradigm to generate objects with PBR materials

**[Question]:** {question}
**[Answer]:** {response}
**[Your output]:**

Answer Scoring Prompt

You are an AI assistant who will help me to judge whether two answers are consistent.

**Input Illustration:**
[Standard Answer] is the standard answer to the question

[Model Answer] is the answer extracted from a model's output to this question.

**Task Illustration:** Determine whether [Standard Answer] and [Model Answer] are consistent.
**Consistent Criteria:**
[Multiple-Choice questions]
1. If the [Model Answer] is the option letter, then it must completely matches the [Standard Answer].
2. If the [Model Answer] is not an option letter, then the [Model Answer] must completely match the option content of [Standard Answer].
[Nan-Multiple-Choice questions]
1. The [Model Answer] and [Standard Answer] should exactly match.
2. If the meaning is expressed in the same way, it is also considered consistent, for example, 0.5m and 50cm.
**Output Format:**
1. If they are consistent, output 1; if they are different, output 0.
2. DIRECTLY output 1 or 0 without any other content.

**Example 1:**
Question: What is the main object in image?
Options: A. teddy bear B. rabbit C. cat D. dog
[Model Answer]: a cute teddy bear
[Standard Answer]: A
Your output: 1

**Example 2:**
Question: Find the value of AB. Choices: A.1; B.5; C.9; D.10
[Model Answer]: 5
[Standard Answer]: B
Your output: 1

**Example 3:**
Question: Three of the following four slides are from the same presentation, but one is from a different one. Please identify the outlier: ¡image¿ ¡image¿ ¡image¿ ¡image¿
[Model Answer]: the forth image
[Standard Answer]: the third image
Your output: 0

**[Question]:** {question}
**[Model Answer]:** {extract answer}
**[Standard Answer]:** {gt answer}
**Your output:**

## F  ERROR ANALYSIS

**Critical Visual Feature Omission**

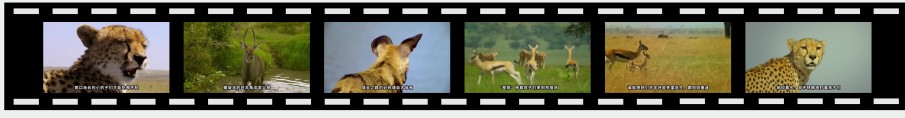

**Question:** Where did the umbrella, used as a weapon by the shop owner, end up?
A: Taken by a masked person.
B: Fell to the ground
C: Put it back on the shelf
D: Stuck in the door

**Answer:** D: Stuck in the door

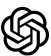
**o1**

Step by step: 1. The shop owner grabs an umbrella to use as a weapon. 2. He attempts to strike the masked individual with it. 3. During the ensuing fight, it is knocked away from his grasp. 4. After that, the umbrella is visibly out of his hands and does not reappear in use. Hence, the umbrella ends up on the ground (Choice B).

**Cot annotation**
1. 0:28-0:33, after the shopkeeper fell, he picked up an umbrella from the ground as a weapon to attack the masked person.
2. 0:33-0:36, during the fight, the umbrella in the shop owner's hand was taken by the masked person.
3. 0:36-0:38, the masked person threw the umbrella in their hand outward, and it stuck in the door.
4. Subsequently, the umbrella did not appear in the footage again, so it can be inferred that the umbrella was finally stuck on the door.

**Excessive inference**

**Question:** What are the reasons for the failure of feral dogs in hunting?
A: Preempted by other predators
B: Let the prey occupy the advantageous terrain
C: Running too slowly.
D: Disturbed by humans

**Answer:** B: Let the prey occupy the advantageous terrain

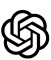
**o1**

Step-by-step reasoning: 1. From the scenes showing African wild dogs (also referred to in the subtitle translation as "青少年野犬"), it's made clear they often operate in open grasslands where multiple predators (such as hyenas or lions) may appear. These other predators can seize the opportunity to steal a kill or drive the wild dogs away. This supports option A: "Preempted by other predators." 2. In the video, certain prey (e.g., waterbuck) are shown seeking refuge in water or dense vegetation, terrains that give them a defensive advantage (especially with sharp horns or hiding spots). The wild dogs often struggle with these less favorable conditions. This corresponds to option B: "Let the prey occupy the advantageous terrain." 3. There is no indication that the wild dogs fail because they "run too slowly." On the contrary, African wild dogs are known for their stamina and speed, so option C does not fit what the footage and narration convey. 4. The footage does not show or discuss human interference disturbing the hunting process, so option D is not supported. Therefore, the primary reasons for failures in feral dog hunts, as per the video, are: • A (preempted by other predators) • B (letting the prey occupy advantageous terrain).

Figure 5: **Common error examples of the model.** The top image shows reasoning failures from missing critical visual features, while the bottom image demonstrates excessive inferences beyond the video content, leading to incorrect answers.

## Incorrect Knowledge Inference

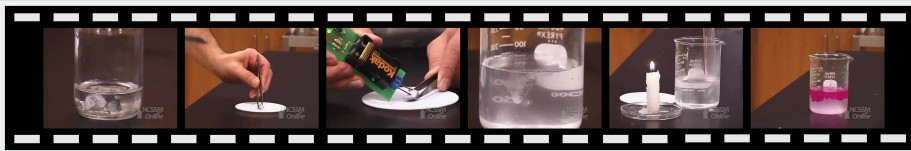

**Question:** Which physical property of the substance is not demonstrated in the video?
A: Conductive
B: Insoluble in oil
C: Silver-white solid
D: Flammable
E: Soft texture

**Answer:** D: Flammable

**Qwen2.5-VL-72B**

To determine which physical attribute of the substance is not demonstrated in the video, let's analyze each option based on the observations from the video: A. **Conductive**: The video shows a piece of the substance being tested with a conductivity probe, indicating that it conducts electricity. B. **Insoluble in oil**: The video does not show any interaction between the substance and oil. It only demonstrates the substance dissolving in water. C. **Silver-white solid**: The substance is shown to be a silver-white solid when it is cut and placed on a plate. D. **Flammable**: The video shows the substance reacting with water, producing hydrogen gas, which is then ignited using a candle, demonstrating its flammability. E. **Soft texture**: The substance is cut with a knife, suggesting that it has a soft texture. Given this analysis, the attribute that is not demonstrated in the video is: **B. Insoluble in oil** Final Answer: B. Insoluble in oil

## Grounding Offset is Significant

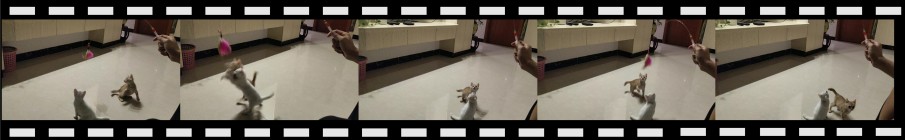

**Question:** For the statement or question 'Which cat shows a slowdown in movement, possibly indicating a loss of interest in the cat toy?', what are the absolute coordinates of the object at 8.9s? Format them as [x1, y1, x2, y2], where [x1, y1] represents the top-left corner and [x2, y2] represents the bottom-right corner.

**Answer:** [1004, 788, 1274, 1079]

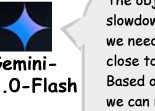

**Gemini-2.0-Flash**

The object at 8.9s is the white cat, which may be showing a slowdown in movement. To determine the absolute coordinates, we need to examine the frame at 00:00:08 (as 8.9s is very close to 8s and frame at 8s is representative of 8.9s). Based on visual inspection of the frame at timestamp 00:00:08, we can manually annotate the bounding box for the white cat. The top-left corner of the white cat appears to be approximately at pixel coordinates [240, 595]. The bottom-right corner of the white cat appears to be approximately at pixel coordinates [410, 750]. Therefore, the absolute coordinates of the white cat at 8.9s are [240, 595, 410, 750]. Final Answer: The final answer is: [240, 595, 410, 750]

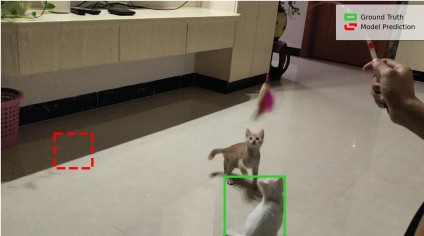

Figure 6: **Common error examples of the model.** The top example shows incorrect reasoning due to the model's lack of domain-specific knowledge, while the bottom one highlights significant deviations caused by poor spatiotemporal localization capabilities.

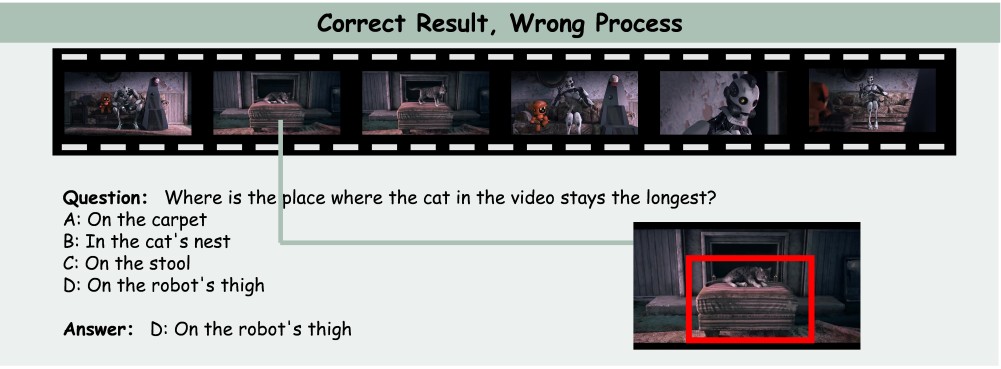

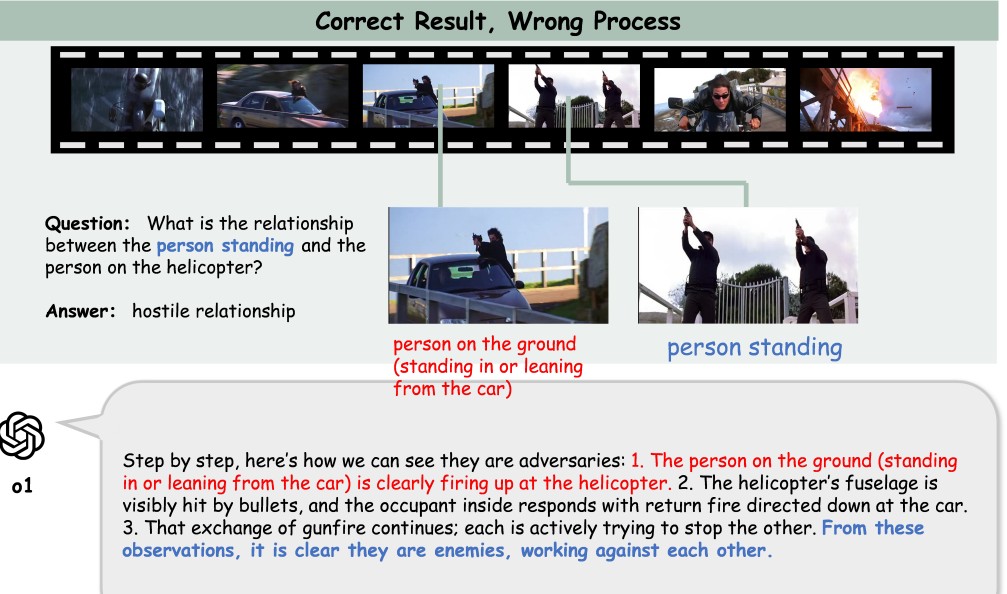

Figure 7: **Common error examples of the model (Correct result, wrong process).** The top example shows the model incorrectly located the initial position of the object (cat) to be identified, while the bottom one shows the model mistakenly identified the person inside the car in the video as the standing person mentioned in the question.

# G MORE QUALITATIVE EXAMPLES

## G.1 OVERVIEW OF EXAMPLES ACROSS DIFFERENT DIMENSIONS

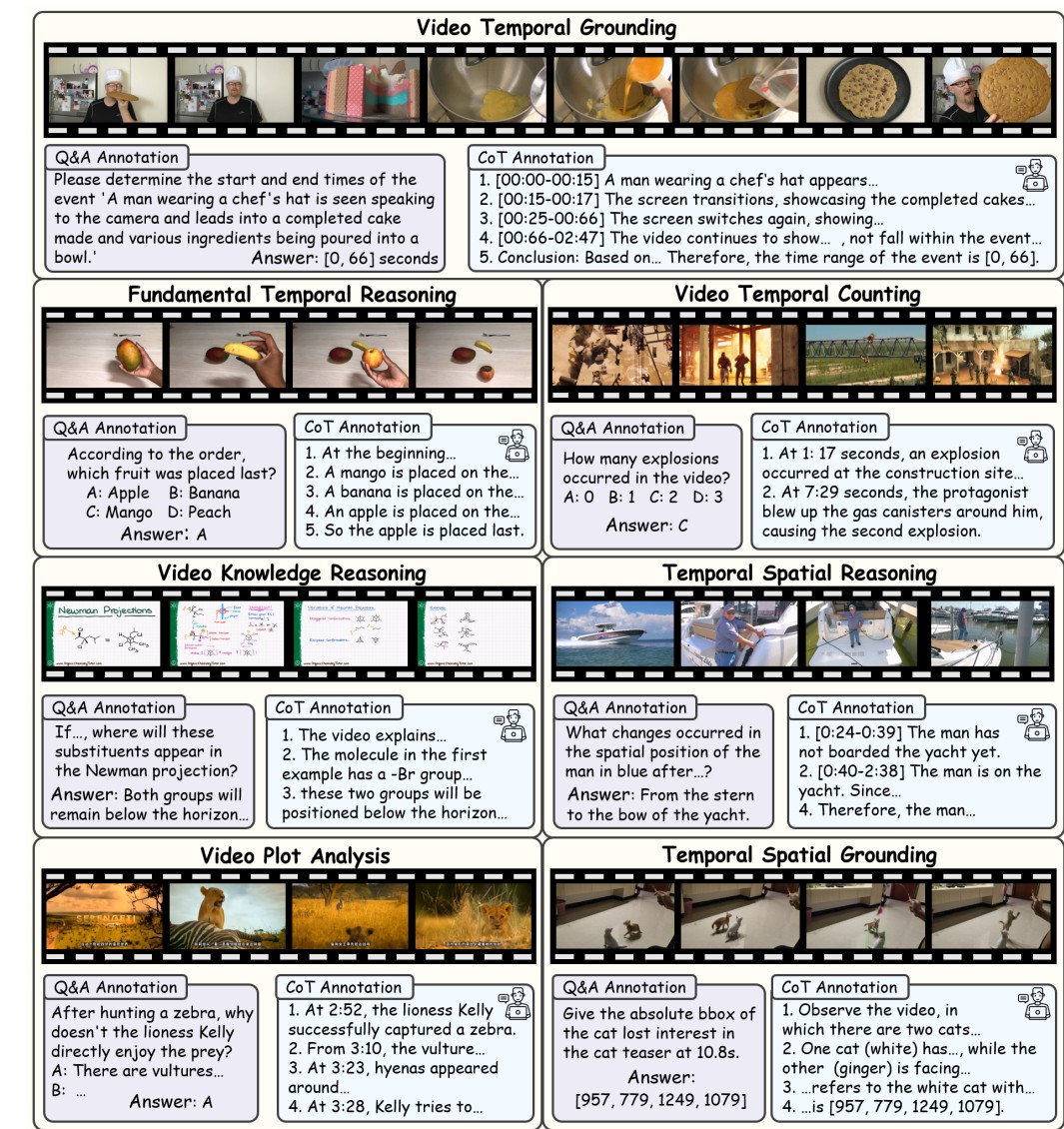

Figure 8: **Cases across dimensions.** VCR-Bench encompasses seven distinct task dimensions spanning multiple competency levels, including spatiotemporal perception, logical reasoning, and knowledge-based analysis.

G.2 SPECIFIC EVALUATION EXAMPLES

---

### Fundamental Temporal Reasoning

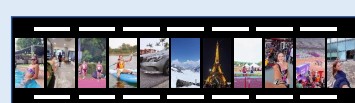

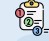 COT Annotation:

1. 0:05 The first display of the running race, labeled "February",
2. 0:08 The second display of the running race scene, labeled "August",
3. Therefore, the lady participated in the running competition for the second time in August.

**Question:** In which month did the woman participate in the running competition for the second time in the video?

**Answer:** August

---

**gemini-2.0-flash-thinking-exp :**

**Prediction:**

Here are the steps to answer the question: Step 1. **Watch the video carefully** and pay attention to the month labels that appear on the screen. Step 2. **Identify the scenes** where the woman is participating in a running competition. Step 3. **Note the months** associated with each running competition scene. Step 4.**Determine the month** of the *second* running competition scene. Based on the video: - At [00:00:04], the month **February** is displayed, and a scene of women running a race is shown. This is the first running competition scene. - At [00:00:07], the month **August** is displayed, and a scene of women jumping hurdles in a race is shown. This is the second running competition scene. Therefore, the woman participated in the running competition for the second time in **August**. **Final Answer:** August

**Precision Evaluation:**

**step_type:** Background Review Steps
**step:** Watch the video carefully and pay attention to the month labels that appear on the screen……
**judgment:** Redundant

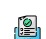 **step_type:** Visual Perception Steps
**step:** At [00:00:04], the month February is displayed, and a scene of women running a race is shown. This is the first running competition scene.
**judgment:** The description of the first running competition scene at 00:00:04 matches the ground truth information.

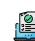 **step_type:** Visual Perception Steps
**step:** At [00:00:07], the month August is displayed, and a scene of women jumping hurdles in a race is shown. This is the second running competition scene.
**judgment:** The description of the second running competition scene at 00:00:07 matches the ground truth information.

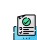 **step_type:** Logical Reasoning Steps
**step:** Therefore, the woman participated in the running competition for the second time in August.
**judgment:** The conclusion that the second participation was in August is directly entailed by the video description steps and matches the ground truth.

### Precision= 3/3

**Recall Evaluation :**

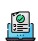 - 0:05 The first display of the running race, labeled "February",

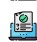 - 0:08 The second display of the running race scene, labeled "August",

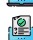 - Therefore, the lady participated in the running competition for the second time in August.

### Recall= 3/3

---

Figure 9: **Examples of FTR.**

## Video Temporal Counting

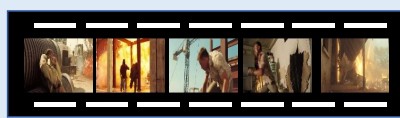

**Question:** How many explosions occurred in the video?

**Choices:** A: 0  B: 1  C: 2  D: 3

**Answer:** C

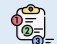 **COT Annotation:**

1. Video 1: 17 seconds, an explosion occurred at the construction site, which is the first explosion appearing in the video.
2. At 7:29 in the video, the protagonist blew up the gas canisters around him, causing the second explosion in the video.
3. Therefore, there were two explosions in total in the video.

**o1:**

**Prediction:**

Step-by-step reasoning: 1. Around 66 seconds, there is a large fiery blast behind some of the characters, clearly visible as an explosion. 2. A few seconds later (around 75 seconds), a second, separate fireball erupts as they continue running. 3. Much later in the video (around 452 seconds), there is another distinct explosion that destroys part of a building. Consequently, there are three clearly identifiable explosions. Answer: D (3).

**Precision Evaluation:**

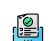
**step_type:** Visual Perception Steps
**step:** Around 66 seconds, there is a large fiery blast behind some of the characters, clearly visible as an explosion.
**judgment:** The video event description time error is more than 3 seconds compared to the ground truth, which states the first explosion occurs at 1:17,

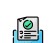
**step_type:** Visual Perception Steps
**step:** A few seconds later (around 75 seconds), a second, separate fireball erupts as they continue running.
**judgment:** This corresponds to the first explosion mentioned in the ground truth.

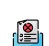
**step_type:** Visual Perception Steps
**step:** Much later in the video (around 452 seconds), there is another distinct explosion that destroys part of a building.
**judgment:** This corresponds to the second explosion mentioned in the ground truth.

**step_type:** Logical Reasoning Steps
**step:** Consequently, there are three clearly identifiable explosions.
**judgment:** The conclusion of three explosions contradicts the ground truth, which only identifies two explosions.

### Precision= 2/4

**Recall Evaluation :**

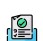
- Video 1: 17 seconds, an explosion occurred at the construction site, which is the first explosion appearing in the video.

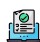
- At 7:29 in the video, the protagonist blew up the gas canisters around him, causing the second explosion in the video.

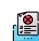
- Therefore, there were two explosions in total in the video.

### Recall= 2/3

Figure 10: **Examples of VTC.**

**Temporal Spatial Grounding**

**Question:** Given the statement or question 'What is the man afraid of being bitten by?', please provide the absolute coordinates of the object referred to at 1.7s in the format [x1, y1, x2, y2], where [x1, y1] represents the top-left corner and [x2, y2] represents the bottom-right corner.

**Answer:** [446, 125, 550, 359]

**COT Annotation:**

1. According to the question description "What is the man afraid of being bitten by?", it can be determined that the question focuses on what the "man" is afraid of being bitten by.
2. Watch the video, there is a man in the center position, holding a stick-like object in his right hand, with a snake in front of his left hand. The man's gaze is directly facing the snake.
3. Based on the problem description and the content of the video, it can be inferred that the man is afraid of being bitten by a snake. Therefore, "What is the man afraid of being bitten by?" refers to "snake."
4. Therefore, the position of the snake at 1.7 seconds is [446, 125, 550, 359].

---

**gemini-2.0-flash-thinking-exp:**

**Prediction:**

The object referred to in the question 'What is the man afraid of being bitten by?' at 1.7s is the snake being held by the man. To provide the absolute coordinates of the snake at 1.7s, we will use the image at 00:00 as a reference since the scene is static and 1.7s falls within the timeframe of these images. We need to identify the bounding box of the snake in the image. By visually inspecting the image at 00:00, we can determine the top-left and bottom-right corners of the snake. The top-left corner of the snake appears to be around the head of the snake, which is located towards the right side and top of the snake figure. Let's estimate the coordinates for the top-left corner. x1: Approximately 445 pixels from the left edge of the image. y1: Approximately 15 pixels from the top edge of the image. The bottom-right corner of the snake appears to be around the tail of the snake, which is located towards the right side and bottom of the snake figure. Let's estimate the coordinates for the bottom-right corner. x2: Approximately 590 pixels from the left edge of the image. y2: Approximately 290 pixels from the top edge of the image. Therefore, the absolute coordinates of the snake are approximately [445, 15, 590, 290]. Final Answer: [445, 15, 590, 290].

**Precision Evaluation:**

**step_type:** Logical Reasoning Steps
**step:** The object referred to in the question 'What is the man afraid of being bitten by?' at 1.7s is the snake being held by the man.
**judgment:** The step correctly identifies the object as the snake based on the question and the video content. This matches the ground truth that the man is afraid of being bitten by a snake.

**step_type:** Visual Perception Steps
**step:** To provide the absolute coordinates of the snake at 1.7s, we will use the image at 00:00 as a reference since the scene is static and 1.7s falls within the timeframe of these images.
**judgment:** The step assumes that the scene is static and uses the image at 00:00 as a reference for 1.7s. This is reasonable and does not contradict the ground truth.

**step_type:** Visual Perception Steps
**step:** The top-left corner of the snake appears to be around the head of the snake, which is located towards the right side and top of the snake figure. Let's estimate the coordinates for the top-left corner. x1: Approximately 445 pixels from the left edge of the image. y1: Approximately 15 pixels from the top edge of the image.
**judgment:** The estimated coordinates for the top-left corner (445, 15) do not match the ground truth coordinates (446, 125). Therefore, this step is incorrect.

**step_type:** Visual Perception Steps
**step:** The bottom-right corner of the snake appears to be around the tail of the snake, which is located towards the right side and bottom of the snake figure. Let's estimate the coordinates for the bottom-right corner. x2: Approximately 590 pixels from the left edge of the image. y2: Approximately 290 pixels from the top edge of the image.
**judgment:** The estimated coordinates for the bottom-right corner (590, 290) do not match the ground truth coordinates (550, 359). Therefore, this step is incorrect.

**step_type:** Logical Reasoning Steps
**step:** Final Answer: [445, 15, 590, 290]
**judgment:** The final answer ([445, 15, 590, 290]) does not match the ground truth coordinates ([446, 125, 550, 359]), IOU=0.36. Therefore, this step is incorrect.

**Precision = 2/5**

**Recall Evaluation**

- According to the question description "What is the man afraid of being bitten by?", it can be determined that the question focuses on what the "man" is afraid of being bitten by.

- Watch the video, there is a man in the center position, holding a stick-like object in his right hand, with a snake in front of his left hand. The man's gaze is directly facing the snake.

- Based on the problem description and the content of the video, it can be inferred that the man is afraid of being bitten by a snake. Therefore, "What is the man afraid of being bitten by?" refers to "snake."

- Therefore, the position of the snake at 1.7 seconds is [446, 125, 550, 359].

**Recall = 2/4**

Figure 11: **Examples of TSG.**

## Video Plot Analysis

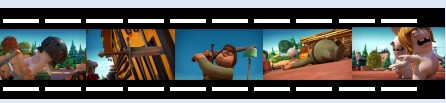

**Question:** How do rabbits defeat their enemies?

**Choices：** A: Received help from the robot
B: Received help from a companion
C: Suddenly awakened superpowers
D: Suddenly acquired a new weapon

**Answer:** B

**COT Annotation:**

1. 0:08-0:12, the rabbit is knocked to the ground by the enemy, and the enemy raises a weapon to finish off the rabbit.
2. 0:13-0:19, suddenly a camera switch, the enemy is knocked to the ground.
3. 0:20-0:23, it shows another rabbit throwing a weapon in the distance.
4. Therefore, it can be inferred that the reason the rabbit is able to defeat its enemies is because it received help from its companions.

---

**Qwen2.5-VL-72B:**

**Prediction:**

To determine how the rabbits defeat their enemies, let's analyze the sequence of events in the video: 1. The video starts with a rabbit character being attacked by a large robot. 2. The rabbit is then shown running and dodging attacks. 3. The rabbit picks up a wooden stick and uses it as a weapon. 4. The rabbit uses the stick to hit the robot, causing it to malfunction and fall apart. Based on this sequence, the rabbit did not receive help from a robot or a companion, nor did it awaken superpowers. Instead, the rabbit acquired a new weapon (the wooden stick) which it used to defeat the enemy. Therefore, the correct answer is: D. Suddenly acquired a new weapon.

**Precision Evaluation:**

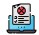
**step_type:** Visual Perception Steps
**step:** The video starts with a rabbit character being attacked by a large robot.
**judgment:** The step describes the initial scene of the video, but the ground truth indicates the rabbit is knocked to the ground by the enemy, not specifically a robot.

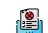
**step_type:** Visual Perception Steps
**step:** The rabbit is then shown running and dodging attacks.
**judgment:** The step describes the rabbit's actions, but there is no scene of running and dodging in the video.

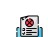
**step_type:** Visual Perception Steps
**step:** The rabbit picks up a wooden stick and uses it as a weapon.
**judgment:** The step describes the rabbit acquiring a weapon, but the ground truth does not mention a wooden stick. The description does not match the ground truth.

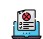
**step_type:** Visual Perception Steps
**step:** The rabbit uses the stick to hit the robot, causing it to malfunction and fall apart.
**judgment:** The step describes the rabbit using a stick to defeat the robot, but the ground truth indicates the enemy is knocked down by another rabbit's action. The description does not match the ground truth.

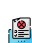
**step_type:** Logical Reasoning Steps
**step:** Based on this sequence, the rabbit did not receive help from a robot or a companion, nor did it awaken superpowers. Instead, the rabbit acquired a new weapon (the wooden stick) which it used to defeat the enemy. Therefore, the correct answer is: D. Suddenly acquired a new weapon.
**judgment:** The conclusion is based on incorrect inferences and descriptions. The ground truth indicates the correct answer is B. Received help from a companion.

### Precision= 0/5

**Recall Evaluation**

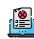
- 0:08-0:12, the rabbit is knocked to the ground by the enemy, and the enemy raises a weapon to finish off the rabbit.

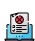
- 0:13-0:19, suddenly a camera switch, the enemy is knocked to the ground.

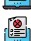
- 0:20-0:23, it shows another rabbit throwing a weapon in the distance.

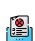
- Therefore, it can be inferred that the reason the rabbit is able to defeat its enemies is because it received help from its companions.

### Recall= 0/4

Figure 12: **Examples of VPA.**

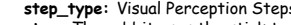

---

### Temporal Spatial Reasoning

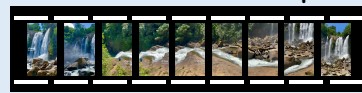

**Question:** What is the position of the person in the video at the waterfall shown at the beginning of the video?

**Answer:** Downstream of the waterfall

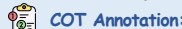 **COT Annotation:**

1. The video begins by showing a panoramic view of the waterfall, with water cascading down from a height, surrounded by rocks and vegetation.
2. The video continuously shows a panoramic view of the waterfall from 0:00 to 0:05.
3. At 0:06, the perspective shifts to a man sitting on a rock.
4. From 0:07 to 0:15 in the video, the camera gradually moves to the right until the full view of the waterfall is shown again, with the man's location being lower than the source of the waterfall.
5. Therefore, it can be determined that it is downstream of the waterfall.

---

**o1:**

**Prediction:**

Step by step: 1. From the start of the video (timestamps around 0–5 s), we see sweeping shots of the waterfall from below, but no person is visible at that time. 2. At about 5.2 s, the perspective shifts to a high vantage point overlooking the waterfall. 3. There we can see a man in a blue shirt. He is seated on the rock ledge right at the top of the waterfall, looking down into the pool below. Thus, the person is positioned at the very top edge of the waterfall, sitting on the rock outcrop.

**Precision Evaluation:**

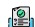 **step_type:** Visual Perception Steps
**step:** The video starts with sweeping shots of the waterfall from below, but no person is visible at that time (timestamps around 0–5 s).
**judgment:** The description matches the ground truth, which states that the video begins with a panoramic view of the waterfall from 0:00 to 0:05, and no person is mentioned as visible during this time.

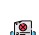 **step_type:** Visual Perception Steps
**step:** At about 5.2 s, the perspective shifts to a high vantage point overlooking the waterfall.
**judgment:** The ground truth states that at 0:06, the perspective shifts to a man sitting on a rock. However, the ground truth also specifies that the man's location is downstream of the waterfall, not a high vantage point overlooking the waterfall.

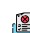 **step_type:** Visual Perception Steps
**step:** There we can see a man in a blue shirt. He is seated on the rock ledge right at the top of the waterfall, looking down into the pool below.
**judgment:** The ground truth states that at 0:06, the perspective shifts to a man sitting on a rock. However, the ground truth also specifies that the man's location is downstream of the waterfall, not at the top edge. This step contradicts the ground truth.

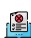 **step_type:** Logical Reasoning Steps
**step:** Thus, the person is positioned at the very top edge of the waterfall, sitting on the rock outcrop.
**judgment:** The conclusion contradicts the ground truth, which states that the man's location is downstream of the waterfall, not at the top edge.

### Precision= 1/4

**Recall Evaluation**

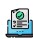 - The video begins by showing a panoramic view of the waterfall, with water cascading down from a height, surrounded by rocks and vegetation.

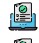 - The video continuously shows a panoramic view of the waterfall from 0:00 to 0:05.

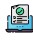 - At 0:06, the perspective shifts to a man sitting on a rock.

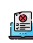 - From 0:07 to 0:15 in the video, the camera gradually moves to the right until the full view of the waterfall is shown again, with the man's location being lower than the source of the waterfall.

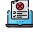 - Therefore, it can be determined that it is downstream of the waterfall.

### Recall= 3/5

Figure 13: **Examples of TSR.**

1728
1729
1730
1731
1732
1733
1734
1735
1736
1737
1738
1739
1740
1741
1742
1743
1744
1745
1746
1747
1748
1749
1750
1751
1752
1753
1754
1755
1756
1757
1758
1759
1760
1761
1762
1763
1764
1765
1766
1767
1768
1769
1770
1771
1772
1773
1774
1775
1776
1777
1778
1779
1780
1781

## Video Knowledge Reasoning

**Question:** Why do light bulbs glow differently when different metal rods are inserted?

**Choices:** A: Different metal rods have different magnetic permeabilities. Metal rods with high magnetic permeability make light bulbs brighter.
B: Different metal rods have different resistances. Metal rods with lower resistance make the bulb brighter.
C: Different metal rods have different magnetic permeabilities. Metal rods with high magnetic permeability make the bulb darker.
D: Different metal rods have different magnetic field strengths. Metal rods with lower magnetic field strengths make the bulb dimmer.

**Answer:** C

**COT Annotation:**

1. From the sixth to the tenth second of the video, the person holds the first metal rod and moves it inside the coil. The light bulb shows no significant change in brightness, and the text displayed above reads "Copper Rod".
2. From the 20th to the 30th second, the second metal rod is moved inside the coil, with the text above showing "Iron Rod." After the iron rod enters the coil, the light bulb goes out. When the rod is removed from the coil, the light bulb lights up again.
3. From the 35th to the 44th second, the fourth rod made of nickel is demonstrated. When the nickel rod enters the coil, the light bulb dims.
4. From the 50th second to the 60th second, an aluminum rod is displayed, and there is no significant change in the brightness of the light bulb.
5. The inductance of an energized solenoid is directly proportional to the magnetic permeability of the inserted metal rod. The higher the magnetic permeability of the metal rod inserted, the greater the inductance, and the greater the resistance of the circuit the bulb is in, and the dimmer the bulb will be. Therefore, the answer is C.

**o1:**

**Prediction:**

Step 1: Observe what is being changed in the video. From the timestamps, we see that the demonstrator is substituting different **metal rods** into the same circuit with a light bulb. Each rod swaps in for the same part of the circuit. Step 2: Notice the effect on the bulb. As each rod is changed, the **brightness** of the bulb changes. Copper and Aluminum rods keep the bulb bright. Iron rod makes the light bulb go out. Step 3: Relate brightness to resistance (Ohm's Law). When a rod has **lower resistance**, current is higher and the bulb glows **brighter**. Conversely, higher resistance yields lower current and the bulb glows **dimmer**. Step 4: Compare with the answer choices. Choices mentioning "magnetic permeability" or "magnetic field strength" do not match the explanation given (the demonstration deals with how well each rod **conducts** electricity, not with its magnetizing effect). The correct explanation is the difference in each rod's **resistance**: rods with **lower resistance** make the bulb brighter and rods with **higher resistance** make it dimmer. Hence, the best matching choice is: (B) Different metal rods have different resistances. Metal rods with lower resistance make the bulb brighter.

**Precision Evaluation:**

**step_type:** Visual Perception Steps
**step:** Observe what is being changed in the video. From the timestamps, we see that the demonstrator is substituting different metal rods into the same circuit with a light bulb. Each rod swaps in for the same part of the circuit.
**judgment:** The step describes the visual observation of the demonstrator substituting metal rods into the circuit. This matches the ground truth information about the demonstrator's actions and the rods being swapped.

**step_type:** Visual Perception Steps
**step:** Notice the effect on the bulb. As each rod is changed, the brightness of the bulb changes. Copper and Aluminum rods keep the bulb bright. Iron rod makes the light bulb go out.
**judgment:** The step describes a relationship between the brightness of the different metal rods. This is consistent with the ground truth, which states that the brightness changes depending on the properties of the rods.

**step_type:** Logical Reasoning Steps
**step:** Relate brightness to resistance (Ohm's Law). When a rod has lower resistance, current is higher and the bulb glows brighter. Conversely, higher resistance yields lower current and the bulb glows dimmer.
**judgment:** The step logically connects brightness to resistance using Ohm's Law. This is consistent with the ground truth, which mentions that resistance affects the brightness of the bulb.

**step_type:** Logical Reasoning Steps
**step:** Compare with the answer choices. Choices mentioning "magnetic permeability" or "magnetic field strength" do not match the explanation given (the demonstration deals with how well each rod conducts electricity, not with its magnetizing effect).
**judgment:** The step eliminates incorrect answer choices based on the explanation provided. However, this contradicts the ground truth, which states that magnetic permeability affects the bulb's brightness. Therefore, this step is incorrect.

**step_type:** Logical Reasoning Steps
**step:** The correct explanation is the difference in each rod's resistance: rods with lower resistance make the bulb brighter and rods with higher resistance make it dimmer. Hence, the best matching choice is: (B)
**judgment:** The final answer is based on the incorrect reasoning that resistance is the determining factor for brightness. The ground truth states that magnetic permeability is the correct explanation, making this step incorrect.

### Precision= 3/5

**Recall Evaluation**

- From the sixth to the tenth second of the video, the person holds the first metal rod and moves it inside the coil. The light bulb shows no significant change in brightness, and the text displayed above reads "Copper Rod".

- From the 20th to the 30th second, the second metal rod is moved inside the coil, with the text above showing "Iron Rod." After the iron rod enters the coil, the light bulb goes out. When the rod is removed from the coil, the light bulb lights up again.

- From the 35th to the 44th second, the fourth rod made of nickel is demonstrated. When the nickel rod enters the coil, the light bulb dims

- From the 50th second to the 60th second, an aluminum rod is displayed, and there is no significant change in the brightness of the light bulb.

- The inductance of an energized solenoid is directly proportional to the magnetic permeability of the inserted metal rod. The higher the magnetic permeability of the metal rod inserted, the greater the inductance, and the greater the resistance of the circuit the bulb is in, and the dimmer the bulb will be. Therefore, the answer is C.

### Recall= 3/5

Figure 14: **Examples of VKR.**

Figure 15: **Examples of VTG.**

# H  USE OF LLMS

Yes. Large language models are used only to assist with language refinement and enhance readability. All technical content, experiments, and analyses are performed entirely by the authors.

