# OpenReview forum: "VCR-Bench: A Comprehensive Evaluation Framework for Video Chain-of-Thought Reasoning"
_ICLR.cc/2026/Conference — Submitted to ICLR 2026_

### Official Review · Reviewer_YcUw · 2025-10-16

**Soundness:** 3
**Presentation:** 3
**Contribution:** 3
**Rating:** 6
**Confidence:** 4

**Summary:**

This paper introduces VCR-Bench, a novel benchmark framework designed to comprehensively evaluate the Chain-of-Thought (CoT) reasoning capabilities of Large Vision-Language Models (LVLMs) for video. The authors argue that existing benchmarks fall short by focusing only on final answer accuracy, failing to assess the reasoning process itself or to distinguish between perception and reasoning failures. VCR-Bench consists of 859 videos and 1,034 high-quality question-answer pairs, each with manually annotated, step-by-step CoT rationales. Each step is tagged as either 'perception' or 'reasoning'. The paper also proposes a "CoT Score" to evaluate the quality of the entire reasoning chain. Extensive experiments on VCR-Bench reveal significant limitations in current LVLMs and pinpoint that the primary bottleneck is in spatio-temporal perception rather than logical reasoning.

**Strengths:**

1.	The work addresses a critical and timely gap in video understanding. By shifting the focus from mere answer accuracy to the quality of the reasoning process, the paper pioneers a more insightful evaluation paradigm that is crucial for understanding and improving advanced LVLMs.
2.	VCR-Bench is comprehensive, built from diverse video sources and covering seven distinct task dimensions. The evaluation method, which decouples perception from reasoning and calculates a CoT score based on recall and precision, is rigorous and provides far deeper insights than traditional metrics.
3.	The paper evaluates a wide range of state-of-the-art models, providing a solid and valuable baseline for the research community. The finding that models struggle more with visual perception than logical reasoning is a significant contribution, as is the strong positive correlation (r=0.89) shown between the CoT score and final accuracy, which validates the framework's effectiveness.

**Weaknesses:**

1.	The paper states that annotations were generated by Gemini 2.0 and then followed by "human verification." More detail on this verification process is needed to fully assess the quality of the ground truth data. For instance, details on the guidelines given to verifiers and the inter-annotator agreement (IAA) scores would significantly strengthen the credibility of the benchmark.
2.	The experiments reveal a near-total failure of all models on the Temporal Spatial Grounding (TSG) task. While the paper attributes this to the need for precise coordinate extraction, a more detailed qualitative error analysis would be beneficial. Understanding how the models fail (e.g., format errors vs. wildly incorrect localization) could provide more targeted insights for future research.
3.	The framework relies entirely on GPT-4o to judge step matches and correctness. This introduces model-dependent bias—if GPT-4o misinterprets reference steps, CoT scores may be inaccurate. The paper acknowledges this limitation but offers no mitigation (e.g., multiple independent annotators, objective matching metrics).

**Questions:**

1.	You mention extracting 64 frames for models without native video support. How did you select these frames (e.g., uniform sampling, keyframe detection)? Did you test if frame selection methods impact model performance, and if so, why did you choose 64 frames specifically?.
2.	Could you elaborate on the human verification stage of your annotation process? Specifically, what guidelines were provided to the human verifiers, and did you measure the inter-annotator agreement to ensure consistency?
3.	The near-zero accuracy on the TSG task is a striking result. Beyond the difficulty of coordinate extraction, did your analysis reveal any common failure patterns? For example, do models tend to identify the correct object but fail at localization, or do they fail at understanding the prompt altogether?

---

> ### Author Response · Authors · 2025-11-20
> **Responses to Official Review by Reviewer YcUw (1/2)**
>
> **We sincerely thank you for the in-depth feedback! We highly value each of your comments, and all your concerns are addressed point by point:**
>
> ### W1 & Q2：Human Verification
> We provide additional details regarding our human verification process:
>
> - All annotators are practitioners in the VLM community and have solid experience with multimodal video understanding and reasoning tasks.
> - We conducted unified training for all annotators to ensure consistent annotation standards.
> - Annotators were instructed to adhere strictly to the objective content presented in the video and to avoid introducing any subjective inference or judgment.
> - We implemented a cross-checking mechanism in which each annotation was reviewed by two to three additional annotators.
> - To ensure high-quality annotations, we excluded any samples that could introduce ambiguity or disagreement.
>
> ### W2 & Q3: Low performance on TSG Task
> Regarding the TSG task, based on our analysis of failure cases (with examples shown in Figure 6 of the appendix), we find that models can generally follow the basic task instructions and produce outputs in the correct format. However, their actual localization performance remains poor. Furthermore, we believe that the deeper reasons may lie in the following aspects:
> 1. The TSG task is still relatively new in multimodal video reasoning, and existing models may lack sufficient training data for it.
> 2. Spatial perception itself is highly challenging for multimodal models. Many models struggle with spatial localization tasks, as also illustrated in the case studies shown in Appendix Figure 6. In our TSG setting, the model must perform not only localization but also additional reasoning to support the localization process, which substantially increases the task difficulty.
>
> Nevertheless, we argue that this task setting effectively evaluates a model’s spatiotemporal perception capabilities and provides valuable insights for model development and performance improvement. As model capabilities continue to advance, the importance of this task will become increasingly evident.
>
> ### W3: GPT-4o to Judge
> Regarding the concern that using a single judge model may introduce bias, this is a common challenge in current LLM evaluation. As reasoning tasks grow more complex and diverse, it is difficult to build rule-based metrics that fully cover all cases, making judge models necessary for open-ended assessment. Notably, mainstream frameworks such as VLMEvalKit and lmms-eval also rely on a single judge model for many of their benchmarks.
>
> Furthermore, to maximize evaluation accuracy, we've established a rigorous assessment protocol that strictly guides GPT-4o through standardized procedures including answer extraction, step decomposition, recall calculation, and precision computation. This structured approach significantly mitigates the impact of GPT-4o's inherent variability. All relevant evaluation prompts are comprehensively documented in the appendix.
>
> Introducing multiple judge models for evaluation is indeed a potentially effective approach. However, it may lead to higher inference costs and inconsistencies across multimodal evaluation results, making it difficult to establish a unified judgment. As a result, this strategy has not yet become mainstream. Here, we also selected several representative models and evaluated their accuracy using different judge models (GPT-4o, Gemini-2.5, o3, and doubao-1.5), as shown in the table below. The results indicate that while there are some differences across judge models, the overall evaluation trends remain consistent, confirming that the results from a single judge model are relatively reliable.
>
> |                    | GPT-4o | Gemini-2.5 | o3   | doubao-1.5 |
> |--------------------|--------|------------|------|------------|
> | Gemini-2.0-Flash   | 51.7   | 52.7       | 53.6 | 51.3       |
> | GPT-4o             | 46.9   | 49.9       | 51.1 | 42.3       |
> | Qwen2.5-VL-7B      | 30.4   | 32.0       | 32.4 | 32.2       |
> | Qwen2.5-VL-72B     | 37.9   | 39.7       | 41.2 | 39.1       |

---

> > ### Author Response · Authors · 2025-11-20
> > **Responses to Official Review by Reviewer YcUw (2/2)**
> >
> > ### Q1: Frame Selection
> > Regarding the selection of video frames, we adopted a uniform frame-sampling strategy. The choice of 64 frames was based on considerations of model performance and inference cost: using too few frames may lead to unreliable predictions, while using too many frames increases inference cost and exceeds the input capacity of some models. Here, we randomly sampled 100 test examples and conducted a frame-number ablation study using Qwen2.5-VL-7B. The results are shown in the table below. As can be observed, the number of frames does have an impact on model performance: with fewer frames, the model’s accuracy drops significantly, whereas after the number of frames reaches a certain level, the performance gains gradually plateau. Based on this balance between inference performance and computational cost, we ultimately selected 64 frames for our experiments.
> >
> > | Frames | 16   | 32   | 64   | 128  |
> > |--------|------|------|------|------|
> > | Accuracy (%) | 13.5 | 33.7 | 36.9 | 37.2 |
> >
> > **The above constitutes our complete response to your comments. Should you have any further questions or require additional clarification, we would be pleased to engage in further discussion.**

---

### Official Review · Reviewer_pcgp · 2025-10-26

**Soundness:** 3
**Presentation:** 3
**Contribution:** 2
**Rating:** 4
**Confidence:** 4

**Summary:**

This paper introduces VCR-Bench, a novel benchmark for evaluating Video Chain-of-Thought (CoT) reasoning in Large Vision-Language Models (LVLMs). The authors correctly identify a critical gap in existing evaluations: current benchmarks typically only assess the accuracy of the final answer, neglecting the intermediate reasoning process. This makes it impossible to determine if a failure stems from flawed perception or flawed logic.

**Strengths:**

1. VCR-Bench provides 1,034 high-quality question-answer pairs over 859 videos. The core contribution is that each sample includes a detailed, manually-annotated, stepwise CoT rationale. Crucially, each step in this rationale is tagged as either a perception or reasoning step, allowing for an unprecedentedly granular analysis.
2. VCR-Bench proposes a novel "CoT score" that measures the precision and recall of generated reasoning steps against reference steps. Experimental results demonstrate a strong correlation between the proposed CoT score and model accuracy.

**Weaknesses:**

1. The benchmark does not evaluate several recently strong multimodal systems (e.g., GPT-5, Gemini-2.5-Pro, Claude 4, and leading open-source models). Including them would better characterize the current frontier and reduce the risk that conclusions reflect outdated model capabilities.
2. Results rely on one judge family. Please assess stronger or different judges (e.g., GPT-5 or dedicated reasoning models such as Grok, plus at least one open-source judge) and report agreement (e.g., κ/ρ) and rank stability across judges.
3. Several takeaways (e.g., perception underperforms reasoning; scaling helps) align with widely reported community observations. To strengthen novelty, consider analyses that go beyond confirmation, such as controlled perturbations (occlusion, motion blur, temporal shuffles), error taxonomy by step type, or causal probes that isolate which perception failures drive end accuracy.

**Questions:**

1. See Weaknesses
2. You report high correlation between CoT score and answer accuracy. Does this imply accuracy alone suffices? Please quantify cases where accuracy is correct but CoT is low (or vice versa), and analyze whether CoT captures diagnostic failures (e.g., missed perception steps) that accuracy alone cannot reveal.

---

> ### Author Response · Authors · 2025-11-20
> **Responses to Official Review by Reviewer pcgp (1/2)**
>
> **We sincerely thank you for the in-depth feedback! We highly value each of your comments, and all your concerns are addressed point by point:**
>
> ### W1：SOTA Model Performance
> Since our work was conducted relatively early and missed some of the latest model results, we supplement the evaluation with updated results of recent models, as shown in the table below. Among them, Gemini-2.0-Flash and Qwen2.5-VL-7B are results already included in the main paper and are shown here for reference. We additionally report the newly updated results for Gemini-2.5-pro, Claude 4, o3, and Qwen3-VL-8B. Overall, these latest models tend to achieve noticeably better performance on VCR-Bench.
> |                     | **Perception** |  |  | **Reasoning** |  |  | **Avg** |  |  |
> |---------------------|-----------|------|------|-----------|------|------|-----------|------|------|
> |                     | precision | recall | f1 | precision | recall | f1 | precision | recall | f1 |
> | **Gemini-2.0-Flash** | 52.1      | 66.6   | 58.5 | 57.4      | 64.6   | 60.8 | 54.0      | 62.1   | 57.7 |
> | **Qwen2.5-VL-7B**    | 31.7      | 53.4   | 39.8 | 34.7      | 37.4   | 36.0 | 33.4      | 44.6   | 38.2 |
> | **Gemini-2.5-pro**   | 74.6      | 62.5   | 68.0 | 70.4      | 67.1   | 68.7 | 71.7      | 63.7   | 67.5 |
> | **Claude 4**         | 58.1      | 46.0   | 51.3 | 51.0      | 53.0   | 52.0 | 60.1      | 46.0   | 52.1 |
> | **o3**               | 66.5      | 53.8   | 59.5 | 56.9      | 59.7   | 58.3 | 67.3      | 55.4   | 60.8 |
> | **Qwen3-VL-8B**      | 62.5      | 41.1   | 49.6 | 50.4      | 44.6   | 47.3 | 61        | 41.8   | 49.6 |
>
> | Model               | FTR  | VTC  | VTG  | VKR  | TSR  | VPA  | TSG  | Avg  |
> |--------------------|------|------|------|------|------|------|------|------|
> | **Gemini-2.0-Flash**   | 66.2 | 51.2 | 62.0 | 64.4 | 54.1 | 58.1 | 4.2  | 51.7 |
> | **Qwen2.5-VL-7B**      | 37.1 | 26.7 | 29.4 | 47.1 | 34.8 | 36.0 | 0.7  | 30.4 |
> | **Gemini-2.5-pro**     | 56.1 | 56.4 | 62.7 | 75.9 | 56.5 | 62.2 | 0.0  | 54.6 |
> | **Claude 4**           | 50.9 | 46.6 | 48.3 | 59.9 | 47.4 | 48.9 | 6.3  | 44.3 |
> | **o3**                 | 52.8 | 45.3 | 49.0 | 61.8 | 44.4 | 53.2 | 4.9  | 44.8 |
> | **Qwen3-VL-8B**        | 37.4 | 37.8 | 47.6 | 54.2 | 29.5 | 32.6 | 3.5  | 35.0 |
>
> ### W2: Different Judges
> We selected several representative models and evaluated the accuracy of their results using different judge models (GPT-4o, Gemini-2.5, o3, and doubao-1.5), as shown in the table below. The table indicates that while there are some differences in the evaluation outcomes across judge models, the overall ranking of the models remains relatively stable, confirming the reliability of using a single judge model for evaluation.
> |                    | GPT-4o | Gemini-2.5 | o3   | doubao-1.5 |
> |--------------------|--------|------------|------|------------|
> | Gemini-2.0-Flash   | 51.7   | 52.7       | 53.6 | 51.3       |
> | GPT-4o             | 46.9   | 49.9       | 51.1 | 42.3       |
> | Qwen2.5-VL-7B      | 30.4   | 32.0       | 32.4 | 32.2       |
> | Qwen2.5-VL-72B     | 37.9   | 39.7       | 41.2 | 39.1       |
>
> ### W3：Strengthen Novelty
> Based on the evaluation results from VCR-Bench, we obtained several conclusions, some of which are consistent with observations previously reported in the community. This demonstrates that our work aligns with the current trends in VLM development and can identify reasoning issues in VLMs through a more systematic evaluation approach, providing stronger experimental support for existing observations, which we believe is valuable.
>
> In addition, we present an analysis of several common error types in Appendix D, including missing key visual cues, over-reasoning, lack of relevant background knowledge, and spatiotemporal perception biases. This error-type analysis further highlights potential reasoning issues in current models.

---

> > ### Author Response · Authors · 2025-11-20
> > **Responses to Official Review by Reviewer pcgp (2/2)**
> >
> > ### Q2: CoT–Accuracy Gap
> > Here, we additionally provide the probability of each model generating a fully correct CoT reasoning process given that the final answer is correct, as shown in the following table.
> >
> > | Model                          | InternVideo2_5_Chat_8B | InternVL2_5-8B | Gemini-1.5-Pro | VideoLLaMA3 | o1   | mPLUG-Owl3-7B | MiniCPM-o2.6-8B | Llama-3.2-11B-Vision | Aria-25B | LLaVA-Video-7B |
> > |-------------------------------|------------------------|----------------|----------------|--------------|------|----------------|------------------|------------------------|-----------|-----------------|
> > | Probability (%)               | 7.9                    | 22.2           | 44.2           | 21.2         | 58.0 | 5.3            | 28.2             | 2.0                    | 12.7       | 11.3             |
> >
> > | Model                          | LLaVA-OneVision-72B | LLaVA-OneVision-7B | Gemini-2.0-Flash | GPT-4o | Claude 3.5 Sonnet | InternVL2.5-78B | Qwen2.5-VL-7B | Qwen2.5-VL-72B | LLaVA-Video-72B |
> > |-------------------------------|----------------------|----------------------|-------------------|--------|--------------------|------------------|----------------|------------------|------------------|
> > | Probability (%)               | 17.6                 | 11.0                  | 56.6              | 49.9   | 52.4               | 24.4             | 40.1           | 53.4             | 18.4             |
> >
> > The results show that even when models arrive at the correct final answer, the probability that their entire reasoning chain is correct remains low. This is partly due to our use of a strict evaluation criterion for CoT reasoning steps, but more importantly, it reflects a common phenomenon in complex video understanding scenarios: models may reach the right answer while still exhibiting gaps or inconsistencies in their reasoning process. We illustrate such cases in Figure 1 and Figure 7 of the appendix. For example, in Figure 1, the video contains two news anchors, but the model overlooks one of them and misclassifies a non-anchor as an anchor, yet still produces the correct answer. This suggests that relying solely on accuracy is insufficient for properly evaluating a model’s performance in video CoT reasoning.
> >
> > **The above constitutes our complete response to your comments. Should you have any further questions or require additional clarification, we would be pleased to engage in further discussion.**

---

> > > ### Comment · Reviewer_pcgp · 2025-11-26
> > >
> > > Thank you for the detailed responses. You have addressed most of my concerns, and the rationale provided for Q2 seems reasonable. Accordingly, I would like to raise my overall evaluation of this paper.

---

> > > > ### Author Response · Authors · 2025-11-26
> > > >
> > > > Thank you so much for the recognition of our responses. We are glad to your positive feedback! Thanks! Following your constructive suggestions, we will make more efforts to improve our paper further. Many thanks for your constructive comments, time, and patience.
> > > >
> > > >
> > > > Best regards and thanks,
> > > >
> > > > Paper 10048 authors

---

### Official Review · Reviewer_Yjo8 · 2025-10-27

**Soundness:** 3
**Presentation:** 3
**Contribution:** 2
**Rating:** 6
**Confidence:** 3

**Summary:**

This paper introduces VCR-Bench, a novel benchmark designed to evaluate the Chain-of-Thought (CoT) reasoning capabilities of Large Vision-Language Models (LVLMs) on video understanding tasks. The authors argue that existing benchmarks are inadequate as they primarily focus on final-answer accuracy, failing to assess the intermediate reasoning process and distinguish between failures in perception versus logical reasoning. VCR-Bench consists of 859 videos and 1,034 high-quality question-answer pairs, spanning seven distinct task dimensions. Crucially, each sample is manually annotated with a detailed, stepwise CoT rationale. The authors propose a new evaluation framework that deconstructs the reasoning process into "Visual Perception" and "Logical Reasoning" steps. It then uses a GPT-4o-based system to calculate recall, precision, and an F1-based "CoT score" by comparing the model's generated reasoning steps against the ground-truth annotations.

**Strengths:**

1. The paper addresses a critical gap in the evaluation of LVLMs. As models become more capable of generating complex, multi-step responses, evaluating only the final answer is insufficient. The motivation, clearly illustrated in Figure 1, is compelling and highlights the need to scrutinize the reasoning process itself.

2. The core contribution—the CoT evaluation strategy—is well-conceived. The deconstruction of reasoning into Visual Perception and Logical Reasoning is an insightful and actionable distinction that allows for a more granular diagnosis of model failures.

3. The paper presents a thorough evaluation across a wide array of 19 models, including top-tier proprietary models and popular open-source alternatives. The findings are highly valuable:

**Weaknesses:**

1. The authors missed the comparison and discussion about the previous sota video reasoning model, e.g. Video-R1[1], LongVILA-R1[2], and the benchmark proposed missed the discussion of solid long video reasoning benchmark - LongVideo-Reason[2]. I suggest the authors add the discussion and analysis of these related works and datasets.

2. The paper is motivated by the phenomenon of models arriving at the correct answer through a flawed process (Figure 1). While the CoT score is designed to detect this, the paper misses an opportunity to explicitly quantify this effect. An analysis showing the distribution of CoT scores for questions that were answered correctly would powerfully reinforce the benchmark's necessity.


[1] Video-R1: Reinforcing Video Reasoning in MLLMs

[2] Scaling RL to Long Videos

**Questions:**

1. Regarding the LLM-as-Judge methodology: Could you provide data on the reliability of using GPT-4o for the recall and precision judgments? Have you performed an analysis of the agreement between GPT-4o's assessments and those of human evaluators on a sample of the results? This would be critical for validating the evaluation pipeline itself.

2. The performance on the Temporal Spatial Grounding (TSG) task is near zero for almost all models. Do you believe this is primarily a failure of the models to generate coordinates in the specified format, or does it reflect a more fundamental inability to link objects in time and space and report their location? Have you experimented with alternative output formats for this task?

---

> ### Author Response · Authors · 2025-11-20
> **Responses to Official Review by Reviewer Yjo8 (1/2)**
>
> **We sincerely thank you for the in-depth feedback! We highly value each of your comments, and all your concerns are addressed point by point:**
>
> ### W1: Sota Video Reasoning Model
> In the table below, we supplement the results of Video-R1-7B and LongVILA-R1-7B, and compare their performance with Qwen2.5-VL-7B. From the results, we can see that compared to Qwen2.5-VL-7B, Video-R1-7B (trained based on Qwen2.5-VL-7B) achieves improved performance on VCR-Bench, particularly in the logic reasoning steps. This confirms that training with RL can enhance the model's capability in video reasoning.
>
> |                     | **Perception** |  |  | **Reasoning** |  |  | **Avg** |  |  |
> |---------------------|-----------|------|------|-----------|------|------|-----------|------|------|
> |                     | precision | recall | f1 | precision | recall | f1 | precision | recall | f1 |
> | **Qwen2.5-VL-7B**    | 31.7      | 53.4   | 39.8 | 34.7      | 37.4   | 36.0 | 33.4      | 44.6   | 38.2 |
> | **Video-R1-7B**      | 35.0      | 20.3   | 25.3 | 50.4      | 59.7   | 54.7 | 38.6      | 41.3   | 39.9 |
> | **LongVILA-R1-7B**   | 39.8      | 18.6   | 25.4 | 50.7      | 28.4   | 36.4 | 39.9      | 20.6   | 27.1 |
>
> | Model               | FTR  | VTC  | VTG  | VKR  | TSR  | VPA  | TSG  | Avg  |
> |--------------------|------|------|------|------|------|------|------|------|
> | **Qwen2.5-VL-7B**      | 37.1 | 26.7 | 29.4 | 47.1 | 34.8 | 36.0 | 0.7  | 30.4 |
> | **Video-R1-7B**        | 32.8 | 34.6 | 22.9 | 47.9 | 46.7 | 24.2 | 1.8  | 33.6 |
> | **LongVILA-R1-7B**     | 33.3 | 36.0 | 17.5 | 41.2 | 28.9 | 30.2 | 0.7  | 27.2 |
>
> ### W2:CoT Scores
> In Figure 4 of our paper, we illustrate the correlation between CoT scores and final answer accuracy, highlighting a strong association between the two. Additionally, we report the probability of each model generating a fully correct CoT reasoning process conditioned on producing the correct final answer, as summarized in the table below.
>
> | Model                          | InternVideo2_5_Chat_8B | InternVL2_5-8B | Gemini-1.5-Pro | VideoLLaMA3 | o1   | mPLUG-Owl3-7B | MiniCPM-o2.6-8B | Llama-3.2-11B-Vision | Aria-25B | LLaVA-Video-7B |
> |-------------------------------|------------------------|----------------|----------------|--------------|------|----------------|------------------|------------------------|-----------|-----------------|
> | Probability (%)               | 7.9                    | 22.2           | 44.2           | 21.2         | 58.0 | 5.3            | 28.2             | 2.0                    | 12.7       | 11.3             |
>
> | Model                          | LLaVA-OneVision-72B | LLaVA-OneVision-7B | Gemini-2.0-Flash | GPT-4o | Claude 3.5 Sonnet | InternVL2.5-78B | Qwen2.5-VL-7B | Qwen2.5-VL-72B | LLaVA-Video-72B |
> |-------------------------------|----------------------|----------------------|-------------------|--------|--------------------|------------------|----------------|------------------|------------------|
> | Probability (%)               | 17.6                 | 11.0                  | 56.6              | 49.9   | 52.4               | 24.4             | 40.1           | 53.4             | 18.4             |
>
> The results show that the probability of models generating fully correct reasoning processes is generally low. This may be partly due to the stringent evaluation criteria applied to the CoT steps, but it also reflects the fact that models tend to produce correct answers while still exhibiting flaws in their reasoning processes when tackling complex video-based questions — as illustrated in Figure 1 of our main paper and Figure 7 in the appendix. This further underscores the necessity of conducting CoT-level process evaluation for the models.

---

> > ### Author Response · Authors · 2025-11-20
> > **Responses to Official Review by Reviewer Yjo8 (2/2)**
> >
> > ### Q1:Regarding the LLM-as-Judge Methodology
> > To verify the validity of GPT-4o evaluations, we selected a subset of samples (100 in total) and compared the results of human evaluation with those of GPT-4o, as shown in the table below. As seen, GPT-4o shows slightly lower scores, indicating some bias. However, given the substantial savings in human labor, we find this bias acceptable.
> >
> >    | Evaluation Type | Precision | Recall | CoT Score (F1) | Average Accuracy |
> >    |-----------------|-----------|--------|-----------|------------------|
> >    | Human           | 52.4      | 55.6   | 54.0      | 45.0             |
> >    | GPT-4o          | 50.3      | 52.4   | 51.3      | 42.8             |
> >
> > In addition, we also report the accuracy results evaluated using other models as judge models (GPT-4o, Gemini-2.5, o3, and doubao-1.5), as shown in the table below. Although the absolute scores differ across judge models, the overall ranking trend remains consistent, indicating that the evaluation results produced by a single judge model are relatively reliable.
> >
> > |                    | GPT-4o | Gemini-2.5 | o3   | doubao-1.5 |
> > |--------------------|--------|------------|------|------------|
> > | Gemini-2.0-Flash   | 51.7   | 52.7       | 53.6 | 51.3       |
> > | GPT-4o             | 46.9   | 49.9       | 51.1 | 42.3       |
> > | Qwen2.5-VL-7B      | 30.4   | 32.0       | 32.4 | 32.2       |
> > | Qwen2.5-VL-72B     | 37.9   | 39.7       | 41.2 | 39.1       |
> >
> > ### Q2: Performance on TSG Task
> > Regarding the performance issues on the TSG task, we believe they may be due to the following two reasons：
> > 1. The TSG task is still relatively new in multimodal video reasoning, and existing models may lack sufficient training data for it.
> > 2. Spatial perception itself is highly challenging for multimodal models. Many models struggle with spatial localization tasks, as also illustrated in the case studies shown in Appendix Figure 6. In our TSG setting, the model must perform not only localization but also additional reasoning to support the localization process, which substantially increases the task difficulty.
> >
> > Moreover, the design of this task was also informed by existing studies[1][2][3]. Currently, spatiotemporal localization in multimodal understanding primarily relies on basic spatial coordinate representations. Although current evaluation results indicate that models are relatively limited in this capability, we believe that this task effectively assesses a model’s spatiotemporal perception and provides valuable guidance for model development and performance improvement. As model capabilities continue to advance, the significance of this task will become increasingly prominent.
> >
> > [1] Qu, Mengxue, et al. "Chatvtg: Video temporal grounding via chat with video dialogue large language models." Proceedings of the IEEE/CVF Conference on Computer Vision and Pattern Recognition. 2024. \
> > [2] Cao, Zhuo, et al. "Flashvtg: Feature layering and adaptive score handling network for video temporal grounding." 2025 IEEE/CVF Winter Conference on Applications of Computer Vision (WACV). IEEE, 2025. \
> > [3] Cheng, Zixu, et al. "V-star: Benchmarking video-llms on video spatio-temporal reasoning." arXiv preprint arXiv:2503.11495 (2025).
> >
> > **The above constitutes our complete response to your comments. Should you have any further questions or require additional clarification, we would be pleased to engage in further discussion.**

---

### Official Review · Reviewer_eAs4 · 2025-10-28

**Soundness:** 2
**Presentation:** 3
**Contribution:** 3
**Rating:** 4
**Confidence:** 5

**Summary:**

This paper focus on the CoT evaluation of Video-LMM.
In detail, this paper comprises 859 videos spanning a variety of video content and durations, along with 1,034 high-quality question-answer pairs, where each pair is manually annotated with a stepwise CoT rationale.
Furthermore, this work evaluates several closed-sourced and open-sourced MLLM, and shows most models score lower on perception than reasoning steps, revealing LVLMs' key bottleneck in temporal-spatial information processing for complex video reasoning.

**Strengths:**

1. The contribution of CoT VideoQA benchmark is meaningful.
2. Experiments on several LMMs shows there're still large room for developing LMMs.

**Weaknesses:**

1. The size of the dataset is limited.
2. For the reasoning process, there may be many possible routes. However, the author cannot handle different reasoning traces.
3. This paper fail to compare with SOTA closed-source and open-source LMM, *e.g.*, Gemini2.5-pro, GPT-5, GPT-o3, Qwen3, etc.
4. The reasoning process is annotated by human, however, humans may have reasoning bias. How does the author handle such bias?
5. I notice the model performance on TSG is extremely low? Could the author provide more analysis on such results? It seems against my observation.

**Questions:**

refer to weaknesses

---

> ### Author Response · Authors · 2025-11-20
> **Responses to Official Review by Reviewer eAs4 (1/2)**
>
> **We sincerely thank you for the in-depth feedback! We highly value each of your comments, and all your concerns are addressed point by point:**
>
> ### W1: Dataset Size
> 1. Our dataset covers seven distinct task dimensions, with approximately 150 QA pairs per dimension. Each QA pair includes complete CoT reasoning steps. While the volume may appear modest compared to some large-scale datasets, we argue that this distribution already encompasses most types of video QA tasks and sufficiently reflects models' true capabilities.
>
> 2. Moreover, the scale of our evaluation set was determined after thorough reference to existing works, such as:
>    - Video-MME (900 videos, 2,700 QA pairs)
>    - LVBench (103 videos, 1,549 QA pairs)
>    - INST-ITBench (206 videos, 1,000 QA pairs)
>    - VideoReferBench (198 videos, 1,000 QA pairs)
>    - MovieChat-1K (130 videos, 1,950 QA pairs)
>
>    A benchmark size of ~1K samples remains highly prevalent in current video QA evaluation frameworks. Therefore, we consider the scale of our dataset to be within a reasonable and well-justified range.
>
>
> ### W2: Different Reasoning Traces
> 1. During data annotation, we took this issue into account. Most samples in the evaluation set have fixed reasoning paths or contain unavoidable key clues. Compared with image data, video data inherently provides a clear, unidirectional temporal dimension, making it easier to define specific key points. This further minimizes the impact of having multiple possible reasoning paths.
> 2. At the same time, we have filtered out data with ambiguous reasoning processes to ensure the reliability of the evaluation as much as possible.
>
> ### W3：SOTA Model Performance
> Since our work was conducted relatively early and missed some of the latest model results, we supplement the evaluation with updated results of recent models, as shown in the table below. Among them, Gemini-2.0-Flash and Qwen2.5-VL-7B are results already included in the main paper and are shown here for reference. We additionally report the newly updated results for Gemini-2.5-pro, Claude 4, o3, and Qwen3-VL-8B. Overall, these latest models tend to achieve noticeably better performance on VCR-Bench.
> |                     | **Perception** |  |  | **Reasoning** |  |  | **Avg** |  |  |
> |---------------------|-----------|------|------|-----------|------|------|-----------|------|------|
> |                     | precision | recall | f1 | precision | recall | f1 | precision | recall | f1 |
> | **Gemini-2.0-Flash** | 52.1      | 66.6   | 58.5 | 57.4      | 64.6   | 60.8 | 54.0      | 62.1   | 57.7 |
> | **Qwen2.5-VL-7B**    | 31.7      | 53.4   | 39.8 | 34.7      | 37.4   | 36.0 | 33.4      | 44.6   | 38.2 |
> | **Gemini-2.5-pro**   | 74.6      | 62.5   | 68.0 | 70.4      | 67.1   | 68.7 | 71.7      | 63.7   | 67.5 |
> | **Claude 4**         | 58.1      | 46.0   | 51.3 | 51.0      | 53.0   | 52.0 | 60.1      | 46.0   | 52.1 |
> | **o3**               | 66.5      | 53.8   | 59.5 | 56.9      | 59.7   | 58.3 | 67.3      | 55.4   | 60.8 |
> | **Qwen3-VL-8B**      | 62.5      | 41.1   | 49.6 | 50.4      | 44.6   | 47.3 | 61        | 41.8   | 49.6 |
>
> | Model               | FTR  | VTC  | VTG  | VKR  | TSR  | VPA  | TSG  | Avg  |
> |--------------------|------|------|------|------|------|------|------|------|
> | **Gemini-2.0-Flash**   | 66.2 | 51.2 | 62.0 | 64.4 | 54.1 | 58.1 | 4.2  | 51.7 |
> | **Qwen2.5-VL-7B**      | 37.1 | 26.7 | 29.4 | 47.1 | 34.8 | 36.0 | 0.7  | 30.4 |
> | **Gemini-2.5-pro**     | 56.1 | 56.4 | 62.7 | 75.9 | 56.5 | 62.2 | 0.0  | 54.6 |
> | **Claude 4**           | 50.9 | 46.6 | 48.3 | 59.9 | 47.4 | 48.9 | 6.3  | 44.3 |
> | **o3**                 | 52.8 | 45.3 | 49.0 | 61.8 | 44.4 | 53.2 | 4.9  | 44.8 |
> | **Qwen3-VL-8B**        | 37.4 | 37.8 | 47.6 | 54.2 | 29.5 | 32.6 | 3.5  | 35.0 |

---

> > ### Author Response · Authors · 2025-11-20
> > **Responses to Official Review by Reviewer eAs4 (2/2)**
> >
> > ### W4：Human Bias
> > We take several measures to minimize potential bias introduced by human annotators. First, all questions and reasoning steps are strictly grounded in the objective visual content of the videos, avoiding any subjective interpretation. Second, all annotators receive systematic training to ensure they fully understand the task requirements. Finally, we adopt a cross-validation workflow in which each annotation is reviewed by 2–3 additional annotators, ensuring that the resulting reasoning processes are as accurate and objective as possible.
> >
> > ### W5: Low Performance on TSG Task
> > Regarding the design of the TSG task and the choice of evaluation metrics, we thoroughly referenced several related works [1][2][3]. The experimental results indeed show that current models perform poorly on the TSG task. We believe this may be due to several factors:
> > 1. The TSG task is still relatively new in multimodal video reasoning, and existing models may lack sufficient training data for it.
> > 2. Spatial perception itself is highly challenging for multimodal models. Many models struggle with spatial localization tasks, as also illustrated in the case studies shown in Appendix Figure 6. In our TSG setting, the model must perform not only localization but also additional reasoning to support the localization process, which substantially increases the task difficulty.
> > Nevertheless, we argue that this task setting effectively evaluates a model’s spatiotemporal perception capabilities and provides valuable insights for model development and performance improvement. As model capabilities continue to advance, the importance of this task will become increasingly evident.
> >
> > [1] Qu, Mengxue, et al. "Chatvtg: Video temporal grounding via chat with video dialogue large language models." Proceedings of the IEEE/CVF Conference on Computer Vision and Pattern Recognition. 2024. \
> > [2] Cao, Zhuo, et al. "Flashvtg: Feature layering and adaptive score handling network for video temporal grounding." 2025 IEEE/CVF Winter Conference on Applications of Computer Vision (WACV). IEEE, 2025. \
> > [3] Cheng, Zixu, et al. "V-star: Benchmarking video-llms on video spatio-temporal reasoning." arXiv preprint arXiv:2503.11495 (2025).
> >
> > **The above constitutes our complete response to your comments. Should you have any further questions or require additional clarification, we would be pleased to engage in further discussion.**

---

> ### Comment · Reviewer_eAs4 · 2025-11-26
>
> The authors address most of my concerns.
> Although I still believe the dataset size should be large for the significant test (for current version, significant test is almost inapplicable in each sub-task for existing method, due to the dataset size), I am willing to increase my rating to weak accept.

---

> > ### Author Response · Authors · 2025-11-26
> >
> > Thank you very much for your recognition of our responses. We are delighted to receive your positive feedback! Following your constructive suggestions, we will make further efforts to improve our paper. Regarding the dataset size, we will also consider exploring more comprehensive data pipelines and enhancing the scalability of the dataset. Once again, thank you for your valuable advice.
> >
> > Best regards and thanks,
> >
> > Paper 10048 authors

---

### Meta-Review · Area_Chair_wR51 · 2026-01-04

**Summary:**

Despite the interesting motivation to evaluate Video Chain-of-Thought reasoning, the paper is rejected due to concerns regarding the robustness and scale of the benchmark. While the authors made significant efforts during the rebuttal, critical issues remain regarding the limited dataset size, which hampers the statistical significance of the results. Furthermore, the reliance on a single proprietary model as the primary judge introduces potential biases that were not fully mitigated, and the near-zero performance on the TSG task raises questions about the task's design validity rather than just model capability.

**Reviewer Concerns:**

The authors successfully addressed concerns regarding the comparison with SOTA models and provided clarification on the human verification process for annotations. However, the concern regarding the small dataset size remains a significant outstanding issue, limiting the benchmark's longevity and statistical power. Additionally, concerns about the reliability of the LLM-as-a-judge framework and the validity of the Temporal Spatial Grounding (TSG) task were not fully resolved to the satisfaction of a top-tier venue standards.

**Reviewer Scores:**

Reviewer eAs4 and pcgp actively participated in the discussion and raised their scores to 6 acknowledging the improvements, though eAs4 explicitly retained reservations about the dataset size. Reviewers Yjo8 and YcUw did not participate in the discussion. They likely would have maintained their borderline scores 6 given that the authors provided data addressing their specific questions, but the overall consensus lacks the strong advocacy required for acceptance.

---

### Decision · Program_Chairs · 2026-01-26

Reject